# Post-Occupancy Evaluation of the Biophilic Design in the Workplace for Health and Wellbeing

**Qinghua Lei [1,2], Stephen Siu Yu Lau [3], Chao Yuan [1] and Yi Qi [3,*]**

[1] Department of Architecture, National University of Singapore, Singapore 117566, Singapore; leiqinghua@u.nus.edu (Q.L.); akiyuan@nus.edu.sg (C.Y.)
[2] Design Research Unit, LWK & PARTNERS (HK) Limited, Hong Kong
[3] Department of Architecture, Shenzhen University, Shenzhen 518060, China; ssylau@hku.hk
[*] Correspondence: qiyi@szu.edu.cn

**Abstract:** There is mounting evidence suggesting that workplace design directly connects with workers' health and wellbeing. Additionally, the personal status of the mind can affect subjective attitudes and feelings towards the environment. In this study, the impacts of biophilic design attributes in offices on workers' health and wellbeing are examined. A new post-occupancy evaluation (POE) questionnaire is developed for evaluating the biophilic design for workplace health and wellbeing. A questionnaire and field observations of two green building offices in Singapore and Shenzhen, China, are performed. The main obtained results are: (i) the questionnaire results show that the workers have a moderately high evaluation of the biophilic attributes in the workplace for improving health and wellbeing; (ii) there are significant differences between the self-reported health and nature relatedness of various ages and genders. Furthermore, the present study provides designers with new weighted biophilic design guidelines, specifically for workplace design practices.

**Keywords:** post-occupancy evaluation; biophilic design; workplace; health and wellbeing

## 1. Introduction

### 1.1. Biophilia and Biophilic Design

The term "biophilia" evolved from human evolution research and was coined to describe humans' inherent affinity for the living things in the natural world [1–3]. It explains why we prefer nature, because it is an instinct deeply rooted in the human brain. Based on the further understanding and the experience-based examination of "biophilia", the "biophilia hypothesis" [1] was first proposed in 1993 to emphasize that the human–nature relationship plays a key role in human brain evolution [4,5] as well as physical [6–9] and psychological health [10–12]. It was the first time that investigators postulated tentative responses to clarify why people love nature and why nature positively affects physical and psychological health based on their proficiencies in various fields [13].

Since the health influences of biophilia are supported by robust empirical evidence [11,14–20], researchers have started to explore how to employ biophilia principles in design practice [6,21–23]. Stephen Kellert (1943–2016) first coined the term for design activity that aimed to "rebuild a positive relationship between the natural environment and human in the modern built environment" as "biophilic design" [24–27]. The innovative approach revealed that biophilia research started to transfer from basic research to practical design application and affected sustainable design strategies. Some scholars summarized and classified the natural design features into biophilic design frameworks to guide design activities [25,28–32].

Recently, designers and investigators with majors in the built environment have directed their attention toward biophilic design [21]. With the health benefits of biophilic design, such projects have recently surged in various environmental design typologies, including commercial [13], healthcare [16,17], and urban designs [18–20]. The workplace is

one of the typologies that attracts the attention of researchers. Scholars who research the relationship between the built environment and health found that the environment does not merely directly or indirectly affect human health, but also affects their work and study performance [33]. Studies have shown that biophilic design benefits workers' health and productivity in an office environment [34–37].

In practice, designers have employed biophilic design attributes or design patterns into design projects and proclaimed such as being biophilic design projects, promoting user wellbeing. Although the importance of biophilic design seems to be well-acknowledged, and some international or regional green building and healthy building standards incorporate biophilic design elements into the rating system, such as WELL building standard version 2 and Singapore Green Mark [38], further research on developing building typology-based biophilic design guidelines and assessment methods is necessary. Additionally, the effectiveness of such design in practical design projects for user wellbeing still requires confirmation. More importantly, building typology-based biophilic design guidelines should be appropriately developed, as they can affect the designer's prioritization of the selection of design attributes in design practice. Concerning the assessment methods from users' viewpoints, post-occupancy evaluation (POE) is one of the most effective approaches. POE is a valuable and validated assessment method to systematically evaluate building design, building performance [39], and occupant satisfaction, health, and wellbeing after the occupancy of buildings [40–42].

### 1.2. Post-Occupancy Evaluation

POE, implemented to assess building function, is a research methodology to effectively evaluate whether the building design and building performance meet the design expectations. From a building design standpoint, the function of POE is a post-occupancy investigation on architectural design. The study results provide feedback to architects, which helps them improve their design strategies in future works. Moreover, from the perspective of building operations, the POE results also provide feedback from occupants to the stakeholders and building managers on workplace biophilic design, since POE is one of the mainstream research methods that can effectively diagnose operational problems [43–45]. From the building performance point of view, POE is a process that evaluates building performance by comparing the assessment results and building criteria. The process of evaluation includes data collection and analysis on the building system. POE is a suitable evaluation approach for assessing whether the green building project could achieve the desired energy-saving effect in the operation process. From users' standpoints, POE focuses on examining whether the designed environment meets users' needs. Additionally, building performance evaluation (BPE) is commonly established based on the POE of building performance and users' satisfaction. Moreover, BPE emphasizes physical properties such as building energy-saving and energy consumption, and takes into account whether the building design affects user behavior and satisfaction. As a result, POE is a combination and improvement of the abovementioned three perspectives. Biophilic design is one of the building design strategies for reestablishing the human–nature connection in the built environment for enhancing health and wellbeing; POE for biophilic design is therefore focused on user experience.

Generally, POE studies are classified into three types based on various research approaches: indicative, investigative, and diagnostic POE [40]. Indicative POE is a crude evaluation methodology exploited to rapidly gather the evaluation data from the key personnel [46,47]. Investigative POE includes scrutiny approaches like end-user questionnaire surveys, interviews, photographic recordings, and field measurements to perform a comprehensive analysis. The diagnostic POE study process requires months or years of comprehensive data collection (see Table 1).

**Table 1.** Post-occupancy evaluation methodologies.

| POE Method | Approaches | Features |
|---|---|---|
| Indicative POE | Interview or distribute questionnaires to key personnel | Rapidly collect evaluation data |
| Investigative POE | End-user surveys, interviews, field measurements, photographic recordings | Apply both occupant feedback and documenting environmental data |
| Diagnostic POE | Months or years of evaluation data collection | A long-term data collection procedure |

### 1.3. Significance of POE in Evaluating Design Impact

Unlike the basic research, the building design practices serve the end-users; in other words, the building design directly affects occupant experiences, therefore "user satisfaction" is critical to building design.

In the field of built environment, POE plays a crucial role in completing the lost link in the sustainable design process. It has helped experts to obtain users' feedback over the last five decades [48,49]. In the existing international green building and healthy building certification systems, POE is included in the certification criteria, such as leadership in energy and environmental design (LEED) [50], the first international green building rating system based on the building research establishment environmental assessment method (BREEAM) [51], the Singapore green mark [52], the green mark for a healthier workplace [52], and the healthy building certification system such as the WELL building certification [53]. Credit requirements mainly focus on the occupant satisfaction survey and building performance documentation. Despite being an indoor environment, quality and health are treated as essential parts of the existing POE for assessing a workplace environment. Researchers have developed analytical methods for biophilic design in the workplace [54]. These mainly include observations and occupant surveys: research documenting and analyzing biophilic features as well as conducting questionnaire surveys or interviews for building occupants. Actually, these are standard research methods of investigative POE; nevertheless, there are no existing POE scales that focus on biophilic design in the workplace. According to the research objectives of this study (i.e., to evaluate the subjective health impacts of biophilic design in the workplace), we need to refer to more well-developed scales from other disciplines (e.g., Environmental Psychology). Finally, a scale that integrates health evaluation and building environment evaluation (i.e., POE) is developed for investigation.

### 1.4. Objectives

The objectives of the present study are summarized as (a) evaluating whether the biophilic design would affect workers' self-reported wellbeing in practical office projects, (b) developing a POE questionnaire for assessing the biophilic design of the workplace, accounting for health and wellbeing, and (c) providing new biophilic design guidelines especially for workplaces. The obtained results from the study can effectively assist researchers and designers to improve office biophilic design practices and decision-making regarding the selection of design attributes.

## 2. Methodology

This study employs investigative POE methods, including field observations and questionnaire surveys of users, because of the certificate limitations to conduct long-term research in both office buildings. In general, there are two parts to the research methods. The first part is field observation. The authors apply nine biophilic attributes to the workplace to identify the features in both offices that match the biophilic attributes in the design framework (see Figure 1). The purpose of the observation is to objectively record and summarize

the biophilic design attributes and the design strategies that designers/architects applied in the actual design practices to achieve an appropriate biophilic design in the workplace.

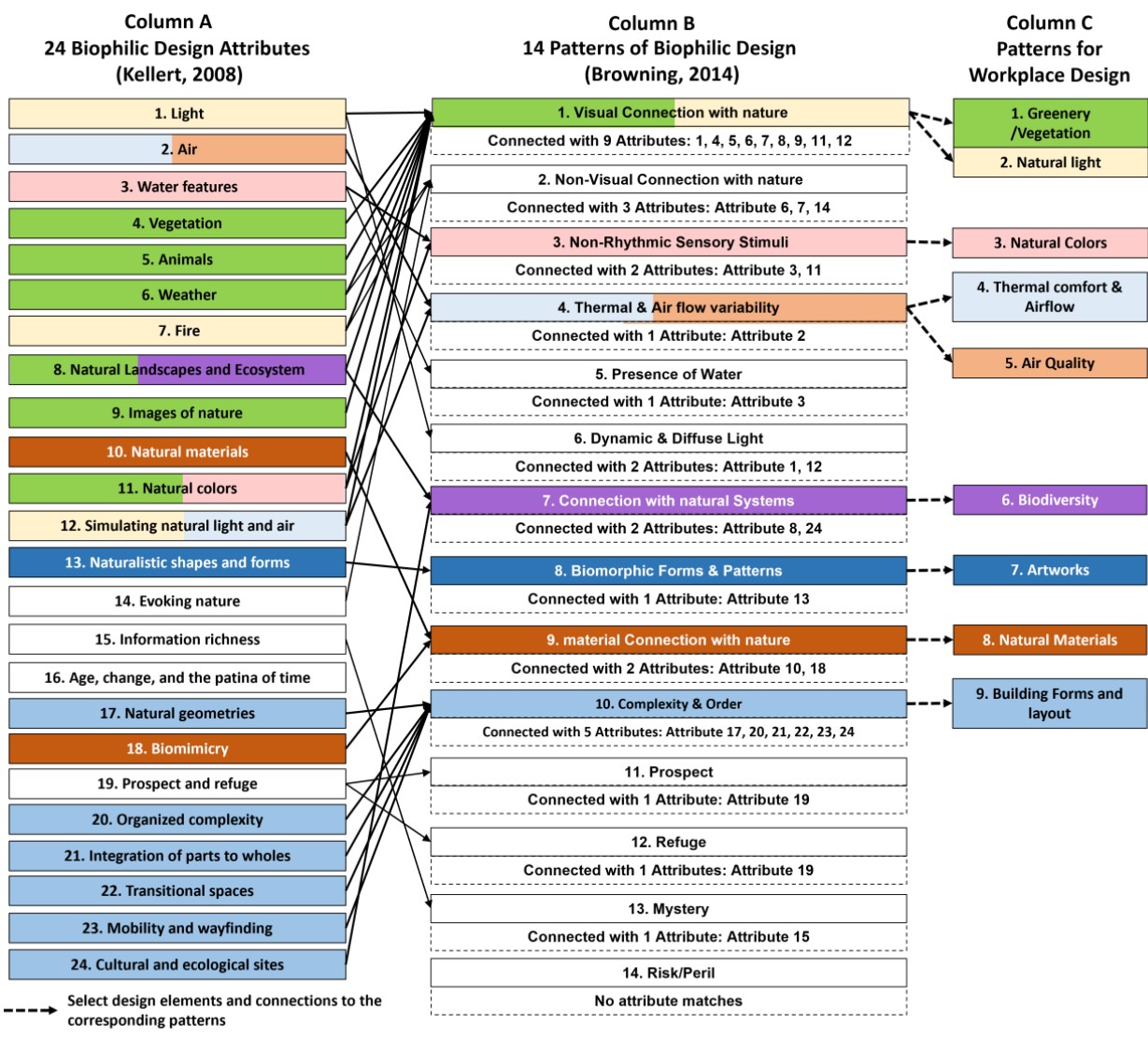

**Figure 1.** The connections between 24 Biophilic Design Attributes (Column A) and 14 Patterns of Biophilic Design (Column B); the nine biophilic design attributes for the workplace (Column C).

Second, the authors establish a self-reported questionnaire to examine the biophilic design attributes of the offices for health and wellbeing. For this purpose, nine biophilic design characteristics from the two mainstream biophilic design frameworks are appropriately reclassified and developed (Figure 1). These nine aspects are included in the last subscale (biophilic design evaluation) of the questionnaire. In the discussion, the authors analyze and weigh the nine points of concern to provide design recommendations for biophilic office design.

### 2.1. Biophilic Design Attributes for Workplace

As a reference to design practices, the validation of design frameworks must be appropriately checked by users. Based on existing biophilic frameworks, two representative frameworks are employed as the references to biophilic design [55]: the 24 Biophilic Design Attributes [25,28] and the 14 Patterns of Biophilic Design [29]. In Kellert's book *Biophilic Design: The Theory, Science, and Practice of Bringing Buildings to Life* [28], the 24 biophilic

design attributes are listed and classified into three types of experience of nature, including "direct experience of nature", "indirect experience of nature", and "experience of place and space" (Column A in Figure 1). Another framework of biophilic design features, the 14 Patterns of Biophilic Design [29], was developed according to the reviews of existing empirical research findings. The 14 biophilic design patterns were divided into 4 levels based on the amount of empirical research. It means that the patterns with more supportive empirical data are marked with more stars, from the lowest score with 0 stars to the highest score with 3 stars (the stars are presented in Column B of Figure 1). These two biophilic frameworks are chosen as research references (definitions of the attributes and patterns are in Appendix A). For the practical project design, evidence shows that instead of applying all of the recommended biophilic design attributes/patterns of a design framework, architects or designers may only choose several attributes/patterns according to various building typologies and functions [55,56]. It implies that, based on the method used to select and develop the biophilic design guidelines for the workplace, the authors should first reclassify and connect the design attributes and patterns of the two selected frameworks according to their definitions. There would also be overlaps between attributes/patterns. For instance, one attribute might match two or more patterns, or one might match two or more attributes. The below graph demonstrates the connections and overlaps of biophilic design attributes/patterns in these two design frameworks (the solid lines link Column A to Column B in Figure 1). As can be observed from the presented connections, 9 of the 24 Biophilic Design Attributes (attributes 1, 4, 5, 6, 7, 8, 9, 11, and 12 in Column A) are interconnected to Pattern 1 of the 14 Patterns of Biophilic Design (Column B); three attributes (attributes 6, 7, and 14 in Column A) are linked to Pattern 2; two attributes (attribute 3 and 11 in Column A) are interrelated with Pattern 3; one attribute (attribute 2 in Column A) is linked to Pattern 4; one attribute (attribute 3 in Column A) is linked to Pattern 5; two attributes (attributes 1 and 12 in Column A) are linked to Pattern 6; two attributes (attributes 8 and 24 in Column A) are linked to Pattern 7; one attribute (attribute 13 in Column A) is linked to Pattern 8; two attributes (attributes 10 and 18) are connected to Pattern 9; five attributes (attributes 17, 20, 21, 22, 23, and 24 in Column A) are linked to Pattern 10; Patterns 11 and 12 are connected to the same attribute: "Attribute 19—Prospect and refuge" in Column A; one attribute (attribute 15 in Column A) is linked to Pattern 13; and there are no attributes that match the final Pattern: "Pattern 15—Risk/Peril".

The authors neglect 7 design patterns from the 14 Patterns of Biophilic Design, which are not representative of the workplace design. For instance, the patterns "Presence of Water", "Prospect", "Refuge", "Mystery", and "Risk" (i.e., itemized patterns 5, 11, 12, 13, and 14 in Column B) are recommended because they are proven to benefit health. However, these patterns are not common in most offices. For instance, in most cases, employers would not create an office environment that makes the workers feel notions of "Prospect", "Refuge", "Mystery", and "Risk". Those patterns are usually applicable in other building typologies, such as hotels or residences. As a result, the authors retain the remaining seven representative design patterns for workplace design (dotted lines and Column C in Figure 1). Finally, the authors specify these design patterns to nine biophilic design attributes. These are "1. Greenery/Vegetation," "2. Natural Light", "3. Natural Colors", "4. Thermal Comfort and Airflow", "5. Air Quality", "6. Biodiversity", "7. Artworks", "8. Natural Materials", and "9. Building Forms and Layout". The recently introduced attributes are based on the definitions and combinations of the patterns/attributes in the two mainstream biophilic design frameworks (Columns A and B).

The 24 Biophilic Design Attributes and the 14 Patterns of Biophilic Design are the two mainstream biophilic designs that are widely applied in practical biophilic design projects. Hence, these two frameworks are the most suitable to be selected as the basis of this experiment.

Furthermore, previous literature-based investigations identified that there are eight factors that affect workers' satisfaction and productivity: "Indoor Air Quality and Ventilation", "Thermal Comfort", "Lighting and Daylighting", "Noise and Acoustics", "Office

Layout", "Biophilia and Views (i.e., biodiversity, greenery, water features)", "Look and Feel (i.e., colors, patterns, spatial settings)", and "Location and Amenities" [54]. Table 2 presents a comparison between the nine selected biophilic design attributes for the workplace in this study and the eight validated factors that affect workers' satisfaction and productivity. There are overlaps between the nine biophilic design attributes and these eight influential factors for the workplace. These overlapped factors highlight the nine biophilic design attributes that are critical to the office design. The validation of the selection of the nine biophilic attributes is demonstrated in the previous literature [54].

**Table 2.** Overlaps between eight physical influential factors for office satisfaction and productivity, and the nine biophilic design attributes for the workplace.

| Eight Physical Influential Factors for Office Satisfaction and Productivity | Overlapping Attributes in the Nine Biophilic Design Attributes for the Workplace |
|---|---|
| Factor 1: Indoor Air Quality and Ventilation | Attribute 4: Thermal Comfort and Airflow; Attribute 5: Air Quality |
| Factor 2: Thermal Comfort | Attribute 4: Thermal Comfort and Airflow |
| Factor 3: Lighting and Daylighting | Attribute 2: Natural Light |
| Factor 4: Noise and Acoustics | N.A. (not included as a biophilic feature) |
| Factor 5: Office Layout | Attribute 9: Building Form and Layout |
| Factor 6: Biophilia and Views (i.e., biodiversity, greenery, water features) | Attribute 1: Greenery/Vegetation; Attribute 6: Biodiversity |
| Factor 7: Look and Feel (i.e., colors, patterns, spatial settings) | Attribute 3: Natural colors; Attribute 7: Artworks; Attribute 8: Natural Materials |
| Factor 8: Location and Amenities | N.A. (not included as a biophilic attribute) |

## 2.2. Evaluation of Biophilic Design Attributes for Workplace by POE

Design Questionnaire for Biophilic Design Evaluation

The questionnaire "Evaluation of the Impacts of Biophilic Design for Workplace Health and Wellbeing" has 5-point scale, with "1" being "Very unsatisfied/Strongly disagree", "5" being "Very satisfied/Strongly agree", and "Neutral" is specified by "3". It means that a value of less than 3 is a negative evaluation, and the values greater than 3 are positive. The questionnaire is provided in the Appendix A.

The major scale of the questionnaire consists of three parts (subscales): general health (GH), nature relatedness (NR), and biophilic design evaluation (BDE). The GH and NR are designed to self-evaluate health status and subjective relatedness to nature. The final part (subscale) is the POE of the office biophilic design. The designation of a set of questions is based on literature reviews. It is worth mentioning that, in this experiment, our questionnaire was developed as a specific questionnaire that focuses on exploring the impacts of biophilic design on office health. The authors did not employ all of the questions given in the reference scales because some of them were unrelated to the biophilic design, office health, and evaluation. Hence, the authors excluded the questions that are unrelated and focus on office health and environment.

The entire questionnaire consists of 27 questions in four sections: demographic information, general health (GH), nature relatedness (NR), and biophilic design evaluation (BDE). In the first section, demographic information is asked, including general questions about age, gender, and educational level. Apart from the basic information, Questions 4 to 7 ask about the work conditions of the respondents as independent variables for analysis, including the weekly working hours, working years, locations of work desks, and daily sedentary time at the work desk (see Table 3).

**Table 3.** Demographic information questions.

| Section | Question Number and Description |
|---|---|
| Demographic information | (1) Gender<br>(2) Age<br>(3) Education level<br>(4) Weekly working hours<br>(5) Daily sedentary time at the work desk<br>(6) Work desk location<br>(7) Working years |

According to the research construct (Table 4), the first subscale is GH. Questions from the World Health Organization Quality of Life (WHOQOL) [57] measure were selected as a basis for those questions targeting general health and wellbeing. The second subscale is NR from the nature relatedness subjective measure scale [58]. The term "nature relatedness" evolved from the biophilia hypothesis, which more explicitly describes the human–nature relationship. Previous research findings demonstrated that individuals who had stronger subjective feelings toward nature had a higher evaluation of their wellbeing [42]. Hence, it is considered the second subscale in the questionnaire for both assessment and inter-correlation analysis. The third subscale is BDE, the POE on the biophilic design features the office environment. The question design is on the basis of the nine biophilic design attributes for the workplace in Figure 1. The questions ask respondents to evaluate the nine biophilic design attributes for health promotion in the workplace. The detailed subscales, assessment items of the questionnaire, and the biophilic design attributes for the workplace in the questionnaire are presented in Table 4.

**Table 4.** Constructs (scales) and assessment items of the questionnaire.

| Scale | Construct | Item Number | Question Number and Description | Matched Biophilic Design Attributes for Workplace |
|---|---|---|---|---|
| Subscale 1 | General Health (GH) | GH 1<br>GH 2<br>GH 3<br>GH 4 | (Q8) Satisfaction with health<br>(Q9) Ability to concentrate<br>(Q10) Satisfaction with work capacity<br>(Q11) Satisfaction with relationships in the workplace | - |
| Subscale 2 | Nature Relatedness (NR) | NR 1<br>NR 2<br>NR 3<br>NR 4 | (Q12) Ideal vacation spot—wilderness area<br>(Q13) Personal actions affect the environment<br>(Q14) Take notice of wildlife<br>(Q15) Personal relationship with nature | - |
| Subscale 3 | Biophilic Design Evaluation (BDE) | BDE 1<br>BDE 2<br>BDE 3<br>BDE 4<br>BDE 5<br>BDE 6<br>BDE 7<br>BDE 8<br>BDE 9<br>BDE 10<br>BDE 11<br>BDE 12 | (Q16) Thermal comfort<br>(Q17) Natural light quality<br>(Q18) Indoor air quality<br>(Q19) Indoor airflow speed<br>(Q20) Spatial arrangement<br>(Q21) Greenery design<br>(Q22) Artwork design<br>(Q23) Biodiversity features<br>(Q24) Material design<br>(Q25) Layout and building shape design<br>(Q26) Color design<br>(Q27) Satisfaction with biophilic features in workplace | Attribute 4<br>Attribute 2<br>Attribute 5<br>Attribute 4<br>Attribute 9<br>Attribute 1<br>Attribute 7<br>Attribute 6<br>Attribute 8<br>Attribute 9<br>Attribute 3<br>- |
| Main Scale | Health and Wellbeing of Biophilic Offices | The sum of the subscales | Question number (8 to 27) | - |

*2.3. Investigate POE Survey*

2.3.1. Selected Offices

The basic information of the sites under investigation is given in Table 5. Office A is in Singapore, known as the "City in the Garden". The selected office is located on the 8th floor. Office B is a representative green office building that was awarded a Three Star Certificate of the Chinese Green Building Label (GBL), Gold Certification of LEED [50] from USGBC, as well as being a winner of many domestic and international awards. The Office B building had been in service for almost ten years, and until the data collection, it was worthwhile to conduct a POE. The building had fourteen floors with twelve floors above ground. The questionnaire covers the 10th-floor office. The two offices have similar features: (1) the urban contexts are similar: both the cities (Singapore and Shenzhen) are typical compact, high density Asian mega-cities; (b) both offices are open-plan.

**Table 5.** Basic information of the studied offices.

| Dimension | Office A | Office B |
|---|---|---|
| Location | Singapore | Shenzhen, South China |
| Climate zone | Tropical monsoon climate | Sub-tropical climate |
| Coordinate | 1°16′ N, 103°5′ E | 22°55′ N, 114.1° E |
| Floor | 8 | 10 |
| Office ventilation type | Central air conditioned | Natural ventilation |
| Temperature in the office | 25 to 26 °C | 26 to 28 °C |
| No. of employees | Approx. 300 | Approx. 150 |

2.3.2. Observation—Biophilic Design Attributes in Selected Offices

The authors implement nine biophilic design attributes for the workplace, displayed in Figure 1, as the reference framework in the observation. Table 6 summarizes the design features that match the attributes for the workplace of the offices under study. First, both offices deploy green features (attribute 1: Greenery/Vegetation). Second, although both offices are air-conditioned, Office B in South China exploits passive ventilation design features, like openable windows for natural ventilation, and semi-open outdoor corridors and spaces (attribute 4: Thermal Comfort and Airflow and attribute 5: Air Quality). Third, Office A in Singapore places more attention on enhanced experiences in the indoor office environment: it employs more interior biophilic design attributes than those of Office B, such as natural materials (attribute 8: Natural Materials) and paintings (attribute 7: Artworks), and has introduced more natural colors (attribute 3: Natural Colors) within the office. The layouts of both offices are designed to put the workstations as close to the windows as possible to achieve natural light or window views (attribute 9: Building Form and Layout). In summary, all nine biophilic design attributes for the workplace are employed in the design practice of the selected offices. The below subsections demonstrate the biophilic design strategies for the four design attributes, which are regularly applicable in both offices.

Attribute 1: Greenery/Vegetation

Plants are one of the biophilic design attributes mostly used at Office B. The plants decorate the whole building in various forms, such as sky gardens, potted plants in workstations, green walls, vertical greening, a roof garden, and green balconies. At both offices, the companies have exploited potted plants in the office interior. Within Office A, some plants have been planted into the office interior, and potted plants placed on the filing cabinets closest to the workstations. Office B regularly provides staff with a potted plant on their work desks. Some employees purchase and place small potted plants on their desk for decoration purposes.

Attribute 4: Thermal Comfort and Airflow and Attribute 5: Air Quality

Office A relies on the air-conditioning system for thermal comfort control because it is located in a tropical climate zone with high daily temperatures over the year. The openable windows of Office B offer a staff-controllable option. In both companies, adjustable curtains are installed on all windows, and Office B is equipped with incorporated panels.

Attribute 8: Natural Materials

The materials used for the building facade and semi-outdoor spaces of Office B, including furniture and flooring, are mainly made of wood and stone. Inside, the interior color is gray, with gray carpets and white ceilings. Natural materials like wooden floors and screens are employed in Office A for interior decoration.

The observation results confirm the efficiency of the design strategies implemented in the offices under study. The anastomosis of the observed and self-reported questionnaire results further supports the effectiveness of biophilic design for health and wellbeing from the users' points of view. The results illustrate that the designers employ various biophilic design strategies to enhance users' experiences. Additionally, the design strategies employed to introduce the natural environment into the office can be diverse.

**Table 6.** Biophilic design attributes in the selected offices and photographic records.

| Biophilic Design Attributes for Workplace | Office A | Office B |
|---|---|---|
| (1) Greenery/Vegetation | 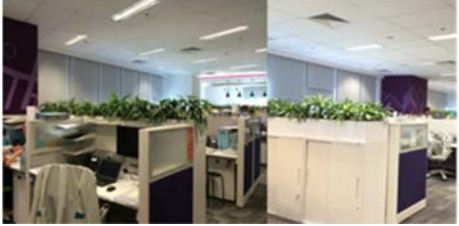 Potted plants, window view of natural scenes | 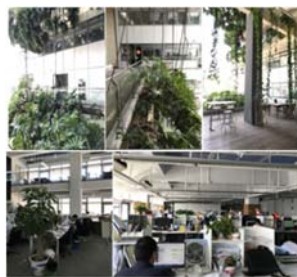 Potted plants, vertical greening, roof garden and green balconies; window view of natural scenes |
| (2) Natural Light | Daylight | Daylight |
| (3) Natural Colors | 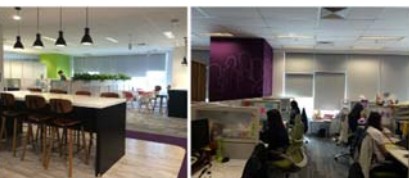 Natural color design | 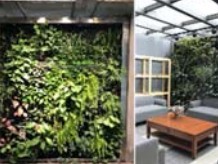 Natural color design |
| (4) Thermal Comfort and Airflow (5) Air Quality | Thermal control (air-conditioning system) | 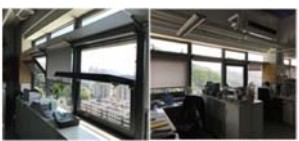 Openable windows for natural ventilation and thermal comfort |

**Table 6.** *Cont.*

| Biophilic Design Attributes for Workplace | Office A | Office B |
|---|---|---|
| (6) Biodiversity |  |  |
| (7) Artworks |   Paintings with natural images and interior biomorphic patterns |   Natural patterns on façade and stairs |
| (8) Natural Materials |   wooden floor and wooden screen |   natural materials |
| (9) Building Form and Layout | Workstations are as close to windows as possible | Workstations are as close to windows as possible, balconies on each floor |

(Photos: by the authors.)

## 3. Questionnaire Results

### 3.1. Demographic Information

A total of 201 valid questionnaires were collected (Table 7), including 161 associated with Office A in Singapore and the others from Office B in South China. Concerning the gender of the respondents, 102 males and 99 females participated (50.7% and 49.3%, respectively). Most of the respondents had a master's degree or above or a bachelor's degree (i.e., 97 persons or 48.3% of the whole). The respondents with secondary school or equivalent degrees were the least, at only seven persons (i.e., 3.5% of the total respondents). A large number of the respondents worked 41 to 50 hours per week (i.e., 138 persons or 68.7%); followed by 34 respondents (16.9%) who worked 50 hours and over, and 29 (14.4%) who worked 30–40 hours per week. A total of 86 (42.8%) respondents reported being sedentary for 5–8 hours per day at their workstation, while 29.4% of them were sedentary for 2–5 hours.

**Table 7.** Demographic information of survey respondents.

| Survey Measures | Items | Number of Persons | Percentage (%) |
|---|---|---|---|
| Gender | Male | 102 | 50.7 |
| | Female | 99 | 49.3 |
| Office Location | Singapore | 161 | 80.1 |
| | Shenzhen, South China | 40 | 19.9 |
| Age | 21–25 | 15 | 7.5 |
| | 26–35 | 84 | 41.8 |
| | 36–45 | 64 | 31.8 |
| | 46–60 | 38 | 18.9 |
| Educational level | Secondary school or equivalent | 7 | 3.5 |
| | Bachelor's degree | 97 | 48.3 |
| | Master's degree or above | 97 | 48.3 |
| Weekly working hours | 30 to 40 h | 29 | 14.4 |
| | 41 to 50 h | 138 | 68.7 |
| | Over 50 h | 34 | 16.9 |
| Daily sedentary time at the work desk | Less than 30 min | 13 | 6.5 |
| | 30 min to 2 h | 19 | 9.5 |
| | 2 to 5 h | 59 | 29.4 |
| | 5 to 8 h | 86 | 42.8 |
| | Above 8 h | 24 | 11.9 |
| Work desk location | Window seats with natural views | 65 | 32.3 |
| | Window seats with urban views | 27 | 13.4 |
| | Aisle seats without window view | 109 | 54.2 |
| Working years | 1 year or less | 47 | 23.4 |
| | 1–3 years | 49 | 24.4 |
| | 3–5 years | 48 | 23.9 |
| | Over 5 years | 57 | 28.4 |

*3.2. Quantitative Results of Impacts of Biophilic Design for Workplace*

The Cronbach's $\alpha$ coefficient value of the main scale is 0.72, while those of the subscales GH, NR, and BDE are respectively obtained as 0.68, 0.79, and 0.63, indicating that the questionnaire is reliable (i.e., an acceptable reliability: Cronbach's $\alpha > 0.6$) [59,60] (Table 8).

**Table 8.** Medians, interquartile range (IQR), and $\alpha$ coefficient values of workers' evaluation based on HWBO, GH, NR, and BDE.

| Structure (Item) | Scale | Median (IQR) | Cronbach's $\alpha$ |
|---|---|---|---|
| Main scale (1) | Health and Wellbeing of Biophilic Offices (HWBO) | 71.00 (8.00) | 0.72 |
| Subscales of main scale (3) | General Health (GH) | 15.00 (2.00) | 0.68 |
| | Nature Relatedness (NR) | 14.00 (3.00) | 0.79 |
| | Biophilic Design Evaluation (BDE) | 42.00 (5.00) | 0.63 |

According to the quantitative results presented in Table 8, the medians (interquartile range—IQR) of the assessment show moderately high opinions toward the health and wellbeing of biophilic offices (HWBO), at 71.00 (8.00) (score range from min. 20 to max. 100). Concerning the self-related evaluation scales of GH and NR (score range from min. 4 to max. 20) the obtained results illustrate moderately high opinions, with values of 15.00 (2.00) and 14.00 (3.00) for GH and NR, respectively. The median (IQR) value of the POE scale BDE is evaluated as 42.00 (19.00) (range of total value: min. 12 to max. 60).

The analysis of the individual items provides more detail of the study responses. As can be seen in the percentage responses for the individual items (Figure 2), most of the responses are distributed in the items "Neutral" and "Agree". The questionnaire results reveal that the employees from the companies under study hold relatively positive opinions on wellbeing, nature relatedness, indoor environmental quality, and biophilic design for their health promotion. At the top of the stacked graph in Figure 2 is the evaluation of satisfaction of the work capacities and relationships in the workplace. About 73.2% of the respondents agree that the workers of the companies under investigation are satisfied with their work capacity (GH3–Q10) and relationships (GH4–Q11). In the subscale of nature relatedness, 62.2% of the respondents agree and 61.7% strongly agree that their actions affect the environment (NR2–Q13), and they take notice of the wildlife in their daily lives (NR3–Q14). Nevertheless, only 47.8% of the respondents selected agree/strongly agree regarding the statement that their ideal spot for a vacation would be a wilderness area (NR1–Q12). Regarding the POE results in the subscale BDE, 63.2% of the workers agree/strongly agree that natural light is an essential biophilic attribute, and that their offices are bright (BDE2–Q17). Furthermore, 60.7% of the workers agree that introducing natural colors into the office benefits workplace health and wellbeing (BDE11–Q26). More than 60% of the respondents believe that greenery is a biophilic design that benefits office wellbeing (BDE6–Q21). Their feedback would be valuable for designers to note that application of the biophilic design attributes in the office design can enhance the experiences and evaluations of workers.

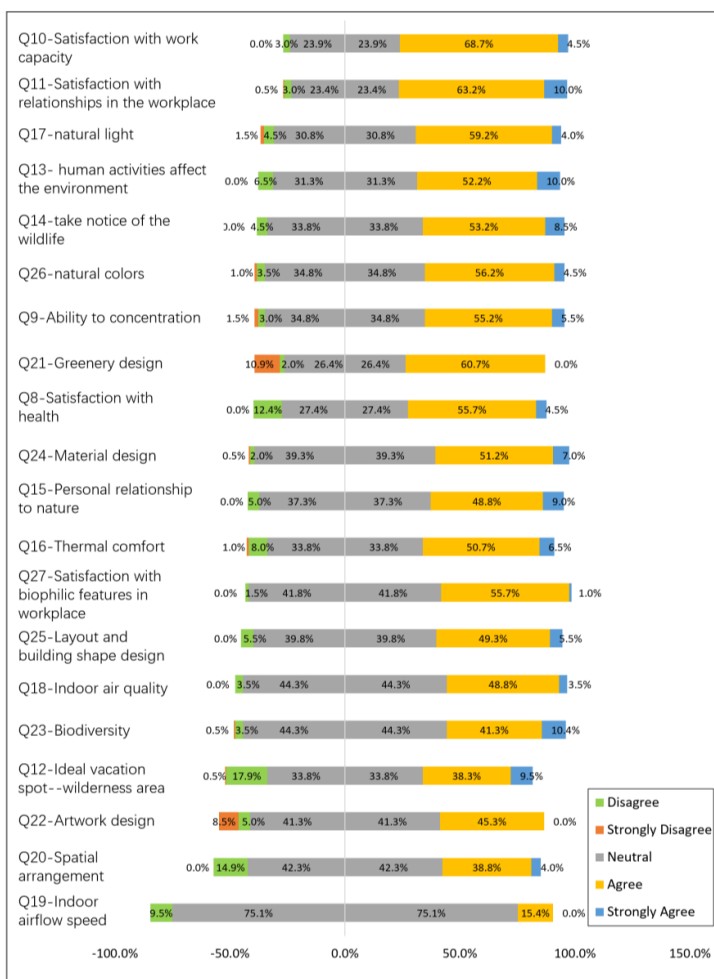

**Figure 2.** Stacked graph with percentage responses for individually arranged items (from the bottom with a high percentage of disagreement to the top with a high percentage of agreement).

The quantitative results of the questionnaire demonstrate that the workers agree that the biophilic design attributes in the office have positive effects on their subjective wellbeing.

## 4. Data Analysis and Discussion

SPSS version 23.0 was exploited for the statistical analysis of the data. Three aspects were analyzed: first, cross-comparisons of different genders, ages, educational levels, weekly working hours, daily sedentary time at work desks, work desk locations, working years, and office locations on the self-reported GH, NR, and BDE. The data did not comply with a normal distribution. Therefore, the authors utilized the independent-samples Kruskal–Wallis test for the comparisons of various groups. Second, Spearman's correlation analysis was utilized to examine the correlation between three subscales. Third, the authors employed the 11 questions (BDE 1 to 11) in the subscale BDE to discuss the weighting of the 9 biophilic design attributes for the workplace design. The weighting was assessed based on the respondents' self-evaluation of the impacts of these attributes on their health. To this end, the weighting was generated by the related samples of Friedman's two-way analysis of variance by ranks.

*4.1. Cross-Comparison of Different Groups of Gender, Age, Educational Levels, Weekly Working Hours, Daily Sedentary Time, Work Desk Locations, Working Years, and Office Locations on Self-Reported GH, NR, and BDE*

The cross-comparison and the pairwise comparison results are presented in Table 9. The results of different genders indicate significant differences in the responses of GH and NR. The male respondents had a higher evaluation of their health ($p = 0.024$ *) and perceived higher personal relatedness to nature ($p = 0.000$ *) compared with the female respondents. Furthermore, the comparisons of the responses of various age ranges reveal an apparent difference in GH ($p = 0.009$ *) and NR ($p = 0.004$ *). The respondents aged 46–60 years reported higher health status (GH) than those aged 36–45 ($p = 0.005$ *). They also reported higher relatedness (NR) with respect to the respondents with ages in the range of 26 to 35 years ($p = 0.012$ *). However, no substantial differences were found in the comparisons of other groups (Educational Levels, Weekly Working Hours, Daily Sedentary Time, Work Desk Locations, Working Years, Office Locations).

**Table 9.** Comparison of independent variables (Gender, Age, Educational Levels, Weekly Working Hours, Daily Sedentary Time, Work Desk Locations, Working Years, Office Locations) on self-reported GH, NR, and BDE.

| Survey Measures | Category | GH | | NR | | BDE | |
|---|---|---|---|---|---|---|---|
| | | Median (IQR) | Sig. | Median (IQR) | Sig. | Median (IQR) | Sig. |
| Gender | Male | 15.00 (2.00) | 0.024 * | 15.00 (2.00) | 0.000 * | 42.00 (5.00) | 0.950 |
| | Female | 15.00 (3.00) | | 14.00 (3.00) | | 42.00 (6.00) | |
| Age | 21–25 | 16.00 (3.00) | 0.009 * | 15.00 (3.00) | 0.004 * | 43.00 (4.00) | 0.115 |
| | 26–35 | 15.00 (2.00) | | 15.00 (3.00) | | 42.00 (5.00) | |
| | 36–45 | 15.00 (4.00) | | 13.50 (3.00) | | 41.00 (6.00) | |
| | 46–60 | 15.50 (2.00) | | 15.00 (2.00) | | 40.50 (6.00) | |
| Educational level | Secondary school or equivalent | 14.00 (4.00) | 0.384 | 14.00 (2.00) | 0.656 | 39.00 (6.00) | 0.391 |
| | Diploma or college certificate | 15.00 (2.00) | | 15.00 (3.00) | | 42.00 (5.00) | |
| | Master's degree or above | 15.00 (2.00) | | 14.00 (3.00) | | 42.00 (5.00) | |

**Table 9.** *Cont.*

| Survey Measures | Category | GH | | NR | | BDE | |
|---|---|---|---|---|---|---|---|
| | | Median (IQR) | Sig. | Median (IQR) | Sig. | Median (IQR) | Sig. |
| Weekly working hours | 30 to 40 h | 14.00 (2.00) | 0.742 | 14.00 (3.00) | 0.445 | 41.00 (6.00) | 0.919 |
| | 41 to 50 h | 15.00 (2.00) | | 14.00 (3.00) | | 42.00 (5.00) | |
| | Over 50 h | 15.00 (3.00) | | 15.00 (3.00) | | 42.00 (5.00) | |
| Daily sedentary time | Less than 30 min | 15.00 (2.00) | 0.549 | 15.00 (2.00) | 0.216 | 42.00 (5.00) | 0.652 |
| | 30 min to 2 h | 14.00 (3.00) | | 13.00 (4.00) | | 43.00 (5.00) | |
| | 2 to 5 h | 15.00 (2.00) | | 14.00 (4.00) | | 42.00 (5.00) | |
| | 5 to 8 h | 15.00 (2.00) | | 15.00 (3.00) | | 41.00 (5.00) | |
| | Over 8 h | 15.00 (2.00) | | 14.50 (4.00) | | 41.00 (7.00) | |
| Work desk location | Window seats with natural views | 15.00 (3.00) | 0.751 | 14.00 (3.00) | 0.228 | 42.00 (5.00) | 0.703 |
| | Window seats with urban views | 16.00 (2.00) | | 14.00 (1.00) | | 42.00 (5.00) | |
| | Aisle seats without window view | 15.00 (2.00) | | 14.00 (3.00) | | 42.00 (5.00) | |
| Working years (in this company) | 1 year or less | 15.00 (2.00) | 0.218 | 15.00 (3.00) | 0.531 | 42.00 (5.00) | 0.458 |
| | 1–3 years | 15.00 (3.00) | | 14.00 (4.00) | | 42.00 (7.00) | |
| | 3–5 years | 15.00 (3.00) | | 14.50 (3.00) | | 42.00 (6.00) | |
| | Over 5 years | 15.00 (2.00) | | 14.00 (3.00) | | 41.00 (6.00) | |
| Office location | Singapore | 15.00 (2.00) | 0.995 | 14.00 (3.00) | 0.701 | 41.00 (5.00) | 0.244 |
| | Shenzhen, China | 15.00 (3.00) | | 15.00 (3.00) | | 43.00 (5.00) | |

* $p < 0.05$: the significance level is 0.05.

### 4.2. Intercorrelation between Three Subscales (GH, NR, BDE)

According to Table 10, Spearman's correlations indicate that nature relatedness (NR) was positively correlated with self-evaluated GH (r = 0.264 **, $p < 0.01$). This result also confirms the previously obtained results that people who had a higher evaluation of nature relatedness also had a higher evaluation of their health. When occupants feel that they have a strong relationship with nature, it is observed that the biophilic environment has positive impacts on their health. More importantly, a significant correlation was also found between biophilic design evaluation and self-reported health (GH), r = 0.270 **, $p < 0.01$, indicating that office biophilic design has positive values on workers' psychological health.

**Table 10.** Intercorrelations between responses of three subscales.

| | GH | NR | BDE |
|---|---|---|---|
| General Health | - | 0.264 ** | 0.270 ** |
| Nature Relatedness | 0.264 ** | - | 0.135 |
| Biophilic Design Evaluation | 0.270 ** | 0.135 | - |

**. Correlation is significant at the 0.01 level (2-tailed).

### 4.3. Weighting of Biophilic Design Attributes for Workplace by POE Results

According to the homogeneous subsets of statistical analysis in Table 11, there are significant differences between the three subsets, while there exist no significant differences within the subsets. The nine biophilic design attributes are weighted and re-arranged into three levels of recommendation (Table 12). Seven attributes, including "Natural Colors", "Natural Light", "Thermal Comfort and Airflow", "Natural Materials", "Greenery", "Biodiversity", and "Air Quality" are included in Level 1. The attributes "Natural Materials", "Greenery", "Biodiversity", "Air Quality", and "Artworks" are rearranged in the second level. Level 3 includes "Biodiversity", "Air Quality", "Artworks", and "Building Form and Layout". There are overlapping attributes in the three levels, but the attribute "Building

Form and Layout" only occurs in Level 3, and "Natural Color", "Natural Light", and "Thermal Comfort and Airflow" are the most recommended attributes that only occur in Level 1.

**Table 11.** Homogeneous subsets of biophilic design attributes for the workplace.

| Biophilic Design Attributes (Arranged from Highest to Lowest from Bottom to Top) | Subset | | |
|---|---|---|---|
| | 1 (Lower Rank) | 2 (Medium Rank) | 3 (Higher Rank) |
| Building Form and Layout | 4.299 | | |
| Artworks | 4.415 | 4.415 | |
| Air Quality | 4.910 | 4.910 | 4.910 |
| Biodiversity | 5.052 | 5.052 | 5.052 |
| Greenery | | 5.167 | 5.167 |
| Natural Materials | | 5.234 | 5.234 |
| Thermal Comfort and Airflow | | | 5.271 |
| Natural Light | | | 5.326 |
| Natural Colors | | | 5.326 |
| Adjusted Sig. (2-sided test) | 0.093 | 0.067 | 0.836 |

Homogeneous subsets are based on asymptotic significances. The significance level is 0.05. Each cell shows the sample average rank.

**Table 12.** Weighting of biophilic design attributes for workplace by POE.

| Level 1 (Higher Rank) | Level 2 (Medium Rank) | Level 3 (Lower Rank) |
|---|---|---|
| Natural Colors | Natural Materials | Biodiversity |
| Natural Light | Greenery | Air Quality |
| Thermal Comfort and Airflow | Biodiversity | Artworks |
| Natural Materials | Air Quality | Building Form and Layout |
| Greenery | Artworks | |
| Biodiversity | | |
| Air Quality | | |

## 5. Conclusions

The health benefits of human exposure to natural attributes have been well studied in previous works [9,11,61–64], and two mainstream biophilic frameworks (the 24 Biophilic Design Attributes and the 14 Patterns of Biophilic Design) for general design have been well populated. Nevertheless, the performed studies and assessments have drawn little attention to exploring new biophilic design frameworks for specific building typologies.

This study highlights the users' preferences and helps to improve decision-making in workplace biophilic design, and enhances the biophilic design's effectiveness. The significant research outputs from the present study are as follows:

(a) The authors developed a POE questionnaire for evaluating the biophilic design for workplace health and wellbeing. The investigation explains that a combined literature review and POE results are one of the practical methodologies to establish biophilic design frameworks for a specific workplace typology. The questionnaire can be applied in future biophilic design research for investigation.

(b) The study provides novel design guidelines for designers with an emphasis on weight for workplace design practices. The weighting results of this study would be especially applicable to the workplace typology. The 14 Patterns of Biophilic Design framework has a broader range of usage for all building typologies, and is more suitable for general design applications. The weighting results of this experiment are not intended to deny the ranking in the 14 Patterns of Biophilic Design. These are exploited to show a new biophilic design framework for the workplace according to the users' points of view (based on the POE results).

(c) The questionnaire results enhance our knowledge of the practical application of biophilic design frameworks for the workplace and contribute to more framework design consideration.

(d) The correlation results support the importance of biophilic design from the users' perspectives. There is a significant correlation between office biophilic design and the self-reported health of employees (r = 0.270 **, *p* < 0.01).

(e) The study results provide designers with evidence-based design attributes for workplace design (i.e., the nine selected workplace biophilic design attributes).

## 6. Limitations and Future Studies

There are limitations in this study. First, the use of only two cases limits the representation of the framework to all workplace biophilic designs due to the small sample size. Further studies could include more offices and locations as experiment samples. Second, the investigative POE studies evaluated the self-reported health (GH), nature relatedness (NR), and biophilic design in the workplace (BDE). The objective of the study was to evaluate the typical biophilic design attributes in the office environment and the correlation between biophilic design and office health. Hence, the research scope is relatively extensive. In future study, the research scope should be narrowed for more intensive investigation.

**Author Contributions:** Conceptualization, Q.L., C.Y. and S.S.Y.L.; methodology, Q.L. and C.Y.; software, Q.L.; validation, S.S.Y.L., C.Y. and Y.Q.; formal analysis, S.S.Y.L., C.Y. and Y.Q.; investigation, Q.L.; resources, S.S.Y.L.; data curation, Q.L.; writing—original draft preparation, Q.L.; writing—review and ed-iting, Q.L., C.Y. and Y.Q.; visualization, Q.L.; supervision, S.S.Y.L., C.Y. and Y.Q.; project ad-ministration, Q.L., S.S.Y.L., C.Y. and Y.Q.; funding acquisition, Y.Q. All authors have read and agreed to the published version of the manuscript.

**Funding:** National Natural Science Foundation of China: 51908360. Industry-Academy Cooperative Education Project from Ministry of Education: 202101126047.

**Institutional Review Board Statement:** Institutional Review Board of National University of Singapore (NUS-IRB); NUS-IRB Ref Code: LS-18-053E.

**Informed Consent Statement:** Informed consent was obtained from all subjects involved in the study.

**Conflicts of Interest:** The authors declare no conflict of interest.

## Appendix A. Definitions of Biophilic Design Attributes and Patterns

| Column | Attributes/Patterns | Definition |
| --- | --- | --- |
| **Column A: 24 Biophilic Design Attributes (Kellert, 2008)** | 1. Light | Glass walls and clerestories, reflecting colors and materials |
| | 2. Air | Natural ventilation |
| | 3. Water features | Views of prominent water bodies, fountains, aquaria, constructed wetlands |
| | 4. Vegetation | Greenery |
| | 5. Animals | Representation of nonhuman animal life |
| | 6. Weather | Views to the outside, operable windows, porches, decks, balconies, colonnades, pavilions, gardens |
| | 7. Fire | Fireplaces and hearths, and simulated by the creative use of light, color, movement, and materials of varying heat conductance |
| | 8. Natural landscapes and ecosystem | Consists of interconnected plants, animals, water, soils, rocks, and geological forms |

| Column | Attributes/Patterns | Definition |
|---|---|---|
| | 9. Images of nature | The image and representation of nature in the built environment—plants, animals, landscapes, water, geological features |
| | 10. Natural materials | Prominent natural building and decorative materials including wood, stone, wool, cotton, and leather, used in a wide array of products, furnishings, fabrics, and other interior and exterior designs |
| | 11. Natural colors | Emphasizes such appealing environmental forms as flowers, sunsets and sunup, rainbows, and certain plants and animals |
| | 12. Simulating natural light and air | - |
| | 13. Naturalistic shapes and forms | The shapes of plants on building facades and columns, animal facsimiles woven into fabrics and coverings |
| | 14. Evoking nature | Draw from design principles and characteristics of the natural world |
| | 15. Information richness | Rich sensory information |
| | 16. Age, change, and the patina of time | Naturally aging materials, weathering, a sense of the passage of time |
| | 17. Natural geometries | Hierarchically organized scales, sinuous rather than rigid artificial geometries, self-repeating but varying patterns |
| | 18. Biomimicry | Forms and functions found in nature, especially among other species |
| | 19. Prospect and refuge | Prospect: long views of surrounding settings Refuge: sites of safety and security |
| | 20. Organized complexity | Complex spaces tend to be variable and diverse, while organized ones possess attributes of connection and coherence |
| | 21. Integration of parts to wholes | Sequential and successional linking of spaces, as well as by clear and discernible boundaries |
| | 22. Transitional spaces | Hallways, thresholds, doorways, gateways, and areas that link the indoors and outdoors especially porches, patios, courtyards, colonnades |
| | 23. Mobility and wayfinding | Clearly understood pathways and points of entry and egress |
| | 24. Cultural and ecological sites | Local landscapes, indigenous flora and fauna, and characteristic meteorological conditions |
| **Column B:** **14 Patterns of Biophilic Design (Browning et al., 2014)** | 1. Visual connection with nature | A view to elements of nature, living systems and natural processes |
| | 2. Non-visual connection with nature | Auditory, haptic, olfactory, or gustatory stimuli that engender a deliberate and positive reference to nature, living systems or natural processes |
| | 3. Non-rhythmic sensory stimuli | Stochastic and ephemeral connections with nature that may be analyzed statistically but may not be predicted precisely |
| | 4. Thermal and air flow variability | Subtle changes in air temperature, relative humidity, air ow across the skin, and surface temperatures that mimic natural environments |

| Column | Attributes/Patterns | Definition |
|---|---|---|
| | 5. Presence of water | A condition that enhances the experience of a place through the seeing, hearing, or touching of water |
| | 6. Dynamic and diffuse light | Leveraging varying intensities of light and shadow that change over time to create conditions that occur in nature |
| | 7. Connection with natural systems | Awareness of natural processes, especially seasonal and temporal changes characteristic of a healthy ecosystem |
| | 8. Biomorphic forms and patterns | Symbolic references to contoured, patterned, textured or numerical arrangements that persist in nature |
| | 9. Material connection with nature | Material and elements from nature that, through minimal processing, reflect the local ecology or geology to create a distinct sense of place |
| | 10. Complexity and order | Adheres to a spatial hierarchy similar to those encountered in nature |
| | 11. Prospect | An unimpeded view over a distance for surveillance and planning |
| | 12. Refuge | A place for withdrawal, from environmental conditions or the main flow of activity, in which the individual is protected from behind and overhead |
| | 13. Mystery | The promise of more information achieved through partially obscured views or other sensory devices that entice the individual to travel deeper into the environment |
| | 14. Risk/Peril | An identifiable threat coupled with a reliable safeguard |

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
