# Peer review of "Post-Occupancy Evaluation of the Biophilic Design in the Workplace for Health and Wellbeing"

_buildings, doi:10.3390/buildings12040417_

Round 1
Reviewer 1 Report
This manuscript presents the results for survey studies carried out in two office buildings in Singapore and China. The method used POE evaluations to determine the impacts of biophilic design on self-reported occupant health and wellbeing, showing possible causal effects as well as some moderating influences across demographics variables. Biophilic design is certainty an area worthy of interest, and the authors’ endeavours along this research line were appreciated and were something, I felt, contributes the knowledge in this domain.
Upon reviewing this manuscript, there were a few areas that I felt could be improved. Firstly, the definitions given to the parameters for biophilic design “cross-over” with those found for indoor environment quality (comment #9). While there is likely some share attributes (e.g. natural light), I felt this aspect of paper could confuse readers not overly familiar with biophilic concepts. The two offices present interesting case studies, but it wasn’t made clear why they were selected and how they served the research aim (comment #12). Explanations here would be welcomed. Finally, please consider revising some of the statistical analyses, which at present make use of parametric assumptions (comments #13 and #17). Although this I don’t believe this would be a large undertaking, the analysis would be better served by non-parametric tests due to the level of measurement used (i.e. 5-point scales). Despite this, I did appreciate the level of detail the analysis went to, and hope that these proposed alterations will benefit the quality of this manuscript. Some further detailed comments are provided below, along with some suggested references for consideration.
Introduction
#1: The definition of biophilia is described as an “inherent love” toward nature. While this is somewhat accurate, it might be more appropriate to elucidate this as an “inherent affinity”.
#2: P1, L39-41: Please provide references to these frameworks. Reading further to page 3, I believe these are the 24 biophilic design attributes [ref. 25,39], and the 14 patterns of biophilic design [ref.40]. Further references around the biophilic concept could also be provided, e.g.:
Bjørn et al., 2009. Biophilia: Does Visual Contact with Nature Impact on Health and Well-Being? International Journal of Environmental Research and Public Health.
Ulrich, 1993. Biophilia, biophobia, and natural landscapes.
Ko et al., 2020. A window view quality assessment framework. LEUKOS.
The latter reference reviewed many international standards that advocate nature and biophilic design for view and building spaces, with examples given to the Singapore context. This somewhat overlaps with my next comment.
#3: P2, 47-53: Although in the past there were few guidelines, nowadays, there may be more standards that focus on nature integration within the built environment. WELL v2 has several features for Nature and Mind, and Biophilia – Parts I and II, with quantitative assessment methods provided. Similarly, the Green Mark system uses the green plot ratio, assigning credits to greenery provision to enhance biodiversity and visual relief. Other standards likely incorporate biophilic elements in building architecture, and could be worth highlighting.
The general issue raised by the authors do not necessarily imply a lack of guidelines for biophilic design, since there are several readily available, but may point toward prioritisation or emphasis of criteria to meet certain varying expectations, which was alluded to on lines 52-53. If the authors agree with this, perhaps this could be revised here to reflect this.
#4: P2, L63: Although I wouldn’t completely rule this out, POE surveys may not always provide feedback to the architect, since they are implemented post design-stage and the building would be operated by facility management or the owner. In my view, POE information had more utility diagnosing operation problems, which can be solved when running the building, identifying prominent sources of dissatisfaction that can prompt action to resolve these issues. Recently POE studies, also using office data, advocate this as benefit to their implementation, albeit not necessarily being the only reason:
Graham et al., 2020. Lessons learned from 20 years of CBE’s occupant surveys. Building & Cities.
Kent et al., 2021. A data-driven analysis of occupant workspace dissatisfaction. Building and Environment.
Cheung et al. 2021. Occupant satisfaction with the indoor environment in seven commercial buildings in Singapore. Building and Environment.
#5: P3, L92: I think refers to “has helped” given the five decades predating this.
#6: P3, L106: While I generally agree with, questions could be raised to whether POE scales should be used to evaluate biophilic design evaluation. Biophilic design is known to elicit mental and physical health benefits, as stated by the authors on page 1, lines 29-32. Therefore, it would be more appropriate to use psychological scales (e.g. PANAS or psychological restoration), instead of design orientated question or survey. If the authors agree with this, this aspect could be revised.
Method
#7: Something I felt would useful would at the beginning would be a clear definition for what “biophilic attributes” refers to. Figure 1 provides some insights into this, but these listed attributes span across different domains and the communal features are not that apparent. This becomes an issue later, since some aspects referring to biophilic design become unclear.
#8: P4, L173-174: The authors state that seven of the patterns from the 14 patterns of biophilic design were discarded. If this was the case, then please better articulate its overarching utility in this study, considering that half of the patterns were not relevant to the research scope.
#9: P4, L180-183: In traditional POE studies and general building science research, daylight, thermal comfort, and air-quality, would be considered as indoor environment parameters (as examples, please see refs. in comment #4), while office layout and building form would be considered a physical and architectural parameters. Reading further to page 5, lines 189-192, the authors begin to suggest to this, but referred to them and others indoor environmental parameters as factors for the workplace. I would suggest better rationalising the connections between the nine design parameters to biophilia to make these more overt.
#10: Figure 1: The image presenting all the linkages is very interesting and is worth emphasising, but contains an overwhelming degree of information, and the text and line sizes are too small for readership. Please consider simplifying the figure. For example, some text boxes many not need further explanation (e.g. presence of water); also the lines connecting column A to the same patterns in column 4 could be colour coordinated.
#11: Table 2: Please consider providing further explanations for this table. It was not clear what the authors wanted to show.
#12: P8, L245: Please specify why these two offices were of interest (e.g., were they comparable or had specify architectural features worthy of study). If possible, please provide more characteristics (e.g. size, floor area, furniture layout (e.g. open-plan or enclosed), etc.) for each office. Later (P10, L299), it says 201 questionnaires were collected, with 161 occupants taking part in the Singaporean office. This led me to believe that this office was much larger than the building studied in China. An image showing the indoor conditions and outdoor façade for each might be beneficial. Many of the explanations found in section 3.1 could be moved into this part of the manuscript, since they many describe and show the existing office conditions and to do necessarily form part of the main results.
Results
#13: Section 3.3. Although I appreciated the thoroughness to which the descriptive statistical was explained, I wasn’t convinced the mean was the best indicator for the data, considering that evaluation scores were collected on a 5-point scale and not a continuous linear one. In-lieu of the mean, please consider using the median and inter-quartile range as the central tendency and dispersion indicators. Figures 1 and 4 can be removed, as the assumption of normality no longer applies (also on P15, L399-400), or replaced with boxplots.
#14: Table 8: Please consider applying benchmarks for what constitute reasonable levels for internal consistency, when using the Cronbach’s Alpha (e.g. α>0.7): Please see, for example: Taber, 2018. The use of Cronbach’s Alpha when developing and reporting research instruments in science education. Research in Science Education.
Tavakol et al. Making sense of Cronbach’s Alpha. International Journal of Medical Education.
#15: Figure 3. The plot is well presented. A few minor notes for improvement: 1) Please consider adding short or abbreviated labels referring to the actual question, instead of codes (e.g. GH3-Q10). This would make it easier for the reader; 2) Round the percentages to the nearest whole number.
#16: P17, L434: Please correct the unfortunate citation error on this line.
#17: P18, section 4.2: Similar to comment #13, the data may be more suited to a Spearman’s correlation test, instead of the Pearson’s correlation coefficient. Due to the reasonable size of the dataset collected, it may not change the interpretation, but would help improve the analytical rigour.
#18: P18, L448-450: Please check whether the sentence is accurate, and correct the table caption numbers; I believe these should be Tables 10 and 11 and 12, not 1, 2 and 3. The sentence reads: Homogenous subsets with significant discrepancies (differences?) across subsets, leading to no significant differences across subsets. The above is not easy to grasp. If the information is accurate, please consider amending this to make this clearer.
Conclusions
#19: While the conclusions were well structured, I felt the authors could have highlighted more the main takeaway messages from their endeavours, in particularly the relationship between biophilic design and occupant health and wellbeing. This seems to be a core aspect of their work, but did really emerge from the final section of their work in the same way it was emphasised in the abstract.
Author Response
Point 1: The definition of biophilia is described as an “inherent love” toward nature. While this is somewhat accurate, it might be more appropriate to elucidate this as an “inherent affinity”.
Response 1: Thank you for your suggestion. The “inherent affinity” is more appropriate, and the term is updated in the manuscript. Revision in Page 1 Line 28:“The term "Biophilia" is evolved from human evolution research and is coined to de-scribe humans' inherent love affinity for the living things in the natural world [1,2].”
Point 2: P1, L39-41: Please provide references to these frameworks. Reading further to page 3, I believe these are the 24 biophilic design attributes [ref. 25,39], and the 14 patterns of biophilic design [ref.40]. Further references around the biophilic concept could also be provided, e.g.:
žBjørn et al., 2009. Biophilia: Does Visual Contact with Nature Impact on Health and Well-Being? International Journal of Environmental Research and Public Health.
žUlrich, 1993. Biophilia, biophobia, and natural landscapes.
žKo et al., 2020. A window view quality assessment framework. LEUKOS.
The latter reference reviewed many international standards that advocate nature and biophilic design for view and building spaces, with examples given to the Singapore context. This somewhat overlaps with my next comment.
Response 2: Thank you for your suggestions. The references are added in the manuscript.
In-text citation in Page 1 Line 44:
“Some scholars summarized and classified the natural design features into biophilic design frameworks to guide design activities [25, 39, 40, 50, 51, 52].”
Three references are added in the References List:
- Ulrich, 1993. Biophilia, biophobia, and natural landscapes. The Biophilia hypothesis. USA: Island Press: Washington, D.C.
- Bjørn et al., 2009. Biophilia: Does Visual Contact with Nature Impact on Health and Well-Being? International Journal of Environmental Research and Public Health. 6, 2332-2343.
- Ko et al., 2021. A Window View Quality Assessment Framework. Leukos. 1-26. DOI: 10.1080/15502724.2021.1965889.
Point 3: P2, 47-53: Although in the past there were few guidelines, nowadays, there may be more standards that focus on nature integration within the built environment. WELL v2 has several features for Nature and Mind, and Biophilia – Parts I and II, with quantitative assessment methods provided. Similarly, the Green Mark system uses the green plot ratio, assigning credits to greenery provision to enhance biodiversity and visual relief. Other standards likely incorporate biophilic elements in building architecture and could be worth highlighting.
The general issue raised by the authors do not necessarily imply a lack of guidelines for biophilic design, since there are several readily available, but may point toward prioritization or emphasis of criteria to meet certain varying expectations, which was alluded to on lines 52-53. If the authors agree with this, perhaps this could be revised here to reflect this.
Response 3: We agree with the comment. There are general biophilic design frameworks (e.g., the 24 Biophilic Design Attributes and the 14 Patterns of Biophilic Design) and green building, healthy building standards (e.g., WELL v2 and Singapore Green Mark) that include biophilia into the certification systems in nowadays.
And the issue raised by the authors is that these biophilic design frameworks (design guidelines or standards) are general design frameworks (design guidelines or standards) which can be applied to all building typologies (i.e., residential buildings, workplaces, retails, etc). Further research needs to be conducted to develop the design guideline specific for workplace. The sentence was rewritten to demonstrate this argument.
Revision in Page 2 Line 56-61:
“Although the importance of biophilic design seems to be well-acknowledged, and some international or regional green building and healthy building standards incorporate biophilic design elements into the rating system, such as WELL building standard version 2 and Singapore Green Mark [53]. However, further research on developing building typology-based biophilic design guidelines and assessment methods are necessary.”
Point 4: P2, L63: Although I wouldn’t completely rule this out, POE surveys may not always provide feedback to the architect, since they are implemented post design-stage and the building would be operated by facility management or the owner. In my view, POE information had more utility diagnosing operation problems, which can be solved when running the building, identifying prominent sources of dissatisfaction that can prompt action to resolve these issues. Recently POE studies, also using office data, advocate this as benefit to their implementation, albeit not necessarily being the only reason:
žGraham et al., 2020. Lessons learned from 20 years of CBE’s occupant surveys. Building & Cities.
žKent et al., 2021. A data-driven analysis of occupant workspace dissatisfaction. Building and Environment.
žCheung et al. 2021. Occupant satisfaction with the indoor environment in seven commercial buildings in Singapore. Building and Environment.
Response 4: We agree with this comment. Apart from the design evaluation, POE is also one of the mainstream research methods that can effectively diagnosing operation problems. The description and citations were inserted in the updated manuscript.
Revision in Page 2 Line 77 ~ 80:
“Moreover, from the perspective of building operation, the POE results also provide evaluation and feedback from occupants to the stakeholders and building managers on workplace biophilic design. Since POE is one of the mainstream research methods that can effectively diagnosing operation problems [54, 55, 56].”
Citations are added in the Reference list:
- Graham, L.T., Parkinson, T., Schiavon, S., 2021. Lessons Learned from 20 years of CBE’s Occupant Surveys. Buildings and Cities 2(1):166-184. DOI: 10.5334/bc.76
- Kent, M., Parkinson, T., Kim, J., Schiavon, S., 2021. A Data-Driven Analysis of Occupant Workspace Dissatisfaction. Building and Environment 205, 108270
- Cheung, T., Schiavon, S., Graham, L.T., Tham, K.W., 2021. Occupant satisfaction with the indoor environment in seven commercial buildings in Singapore. Building and environment (188). DOI:10.1016/j.buildenv.2020.107443
Point 5: P3, L92: I think refers to “has helped” given the five decades predating this.
Response 5: The sentence was rewritten in the updated version.
Revision in Page 3 Line 109:
“It has helped experts to obtain user’s feedbacks over the last five decades [31,32].”
Point 6: P3, L106: While I generally agree with, questions could be raised to whether POE scales should be used to evaluate biophilic design evaluation. Biophilic design is known to elicit mental and physical health benefits, as stated by the authors on page 1, lines 29-32. Therefore, it would be more appropriate to use psychological scales (e.g., PANAS or psychological restoration), instead of design orientated question or survey. If the authors agree with this, this aspect could be revised.
Response 6: Yes, the existing psychological scales are well-developed, but scales for investigating “workplace biophilic design” had not been developed before. Hence, in this study, we develop a method that focus on evaluating the biophilic design elements in workplace. The major scale of the questionnaire consists of three parts (subscales): general health (GH), nature relatedness (NR), and biophilic design evaluation (BDE). The questions in the first and the second subscales are referred to the validated scales--The World Health Organization Quality of Life (WHOQOL) and the nature relatedness (NR). The third scale is focus on evaluation on the biophilic design elements. Hence, the questions in the final section are designed based on the selection of the biophilic design elements/attributes that typically applied in the office design, which are not mentioned in the previous scales.
Point 7: Something I felt would useful would at the beginning would be a clear definition for what “biophilic attributes” refers to. Figure 1 provides some insights into this, but these listed attributes span across different domains and the communal features are not that apparent. This becomes an issue later, since some aspects referring to biophilic design become unclear.
Response 7: The method of selection of the biophilic design attributes/patterns for workplace is:
ŸStep one, find out the correlated biophilic design characteristics from the two mainstream biophilic design frameworks.
ŸStep two, we neglect the patterns which are not representative in office environment (please find the detailed explanations in the respond for Comment #8) and specify the selected design patterns to nine biophilic design attributes. The detailed process is shown in Section 2.1 and Figure 1.
ŸStep three, verify the selection of the nine biophilic attributes matches the validated eight factors that affect workers’ satisfaction and productivity (please find the detailed explanations in the respond for Comment #11).
In terms of the reviewer think that the listed attributes span across different domains and the communal features are not that apparent, we believe it is due to the research perspective of biophilic design is different from the perspectives of building science and traditional POE studies (please find the detailed explanations in the respond for Comment #9).
Point 8: P4, L173-174: The authors state that seven of the patterns from the 14 patterns of biophilic design were discarded. If this was the case, then please better articulate its overarching utility in this study, considering that half of the patterns were not relevant to the research scope.
Response 8: The authors gave the reason after the sentence in Page 4, Line 173-174 of the original manuscript “Second, the authors neglect seven design patterns from the 14 Patterns of Biophilic Design which are not representative of the workplace design.” to explain why the seven of the patterns from the 14 patterns of biophilic design were discarded:
For instance, the patterns “Presence of Water”, “Prospect”, “Refuge”, “Mystery”, and “Risk” (i.e., itemized patterns 5, 11, 12, 13, and 14 in Column B) are recommended but not demonstrative in this building typology (workplace). Those are usually applicable in other building typologies, such as hotels or residential.
To further explain why these patterns are recommended but are discarded: 1) First, these design patterns (i.e., the seven discarded patterns of the 14 biophilic design patterns) are recommended because they are proofed that benefits health. 2) However, these patterns are not common in most offices. For instance, in most cases, the employers would not create an office environment that makes the workers feel “Prospect”, “Refuge”, “Mystery”, and “Risk”. 3) Therefore, we only included those biophilic design patterns that relatively easy to apply in the workplace (e.g., greenery, natural light, artworks), and discard those which are not representative in an office design.
And the sentences are revised to further explain the discard of the seven patterns. Please see the revised contents in the updated manuscript below:
Revision in Page 5 Line 191-197:
“For instance, the patterns “Presence of Water”, “Prospect”, “Refuge”, “Mystery”, and “Risk” (i.e., itemized patterns 5, 11, 12, 13, and 14 in Column B) are recommended because they are proofed that benefits health. However, these patterns are not common in most offices. For instance, in most cases, the employers would not create an office environment that makes the workers feel “Prospect”, “Refuge”, “Mystery”, and “Risk”.”
Point 9: P4, L180-183: In traditional POE studies and general building science research, daylight, thermal comfort, and air-quality, would be considered as indoor environment parameters (as examples, please see refs. in comment #4), while office layout and building form would be considered a physical and architectural parameters. Reading further to page 5, lines 189-192, the authors begin to suggest to this, but referred to them and others indoor environmental parameters as factors for the workplace. I would suggest better rationalizing the connections between the nine design parameters to biophilia to make these more overt.
Response 9: We believe that there is no conflict between the different classifications. The same parameters (e.g., daylight, thermal comfort, and air-quality, office layout and building form) can be classified in different classifications (i.e., the traditional POE frameworks and the biophilic design frameworks) by different perspectives.
1) From perspective of building science, building performance, and traditional POE, these parameters (e.g., daylight, thermal comfort, and air-quality) are considered as indoor environment parameters, and office layout and building form are considered a physical and architectural parameters.
2) On the other hand, from the perspective of biophilia and biophilic design, (e.g., factors workers’ satisfaction and productivity), these parameters are re-classified and defined as the factors affecting workplace health.
Both classifications validated by previous literatures.
Point 10: Figure 1: The image presenting all the linkages is very interesting and is worth emphasizing, but contains an overwhelming degree of information, and the text and line sizes are too small for readership. Please consider simplifying the figure. For example, some text boxes many do not need further explanation (e.g., presence of water); also, the lines connecting column A to the same patterns in column 4 could be color coordinated.
Response 10: Thank you for your comment. We revised the Figure 1. The connections between 24 Biophilic Design Attributes (Column A) and 14 Patterns of Biophilic Design (Column B); The nine biophilic design attributes for the workplace (Column C).
First, to simplify the image, we put all the definitions into a new table in the appendix (Appendix A. Definitions of biophilic design attributes and patterns). In case that there are some readers are not familiar with the biophilic design attributes or patterns:
Revision in Page 4 Line 164:
“These two biophilic frameworks are chosen as research references (definitions of the at-tributes and patterns are in Appendix A).”
Appendix A in Page 22:
“Appendix A. Definitions of biophilic design attributes and patterns”
|
Column |
Attributes/ Patterns |
Definition |
|
Column A: 24 Biophilic Design Attributes (Kellert, 2008) |
1. Light |
Glass walls and clerestories, reflecting colors and materials |
|
2. Air |
Natural ventilation |
|
|
3. Water features |
Views of prominent water bodies, fountains, aquaria, constructed wetlands |
|
|
4. Vegetation |
Greenery |
|
|
5. Animals |
Re-presentation of nonhuman animal life |
|
|
6. Weather |
Views to the outside, operable windows, porches, decks, balconies, colonnades, pavilions, gardens |
|
|
7. Fire |
fireplaces and hearths, and simulated by the creative use of light, color, movement, and materials of varying heat conductance |
|
|
8. Natural Landscapes and Ecosystem |
consist of interconnected plants, animals, water, soils, rocks, and geological forms. |
|
|
9. Images of nature |
The image and representation of nature in the built environment—plants, animals, landscapes, water, geological features |
|
|
10. Natural materials |
Prominent natural building and decorative materials include wood, stone, wool, cotton, and leather, used in a wide array of products, furnishings, fabrics, and other interior and exterior designs. |
|
|
11. Natural colors |
emphasize such appealing environmental forms as flowers, sunsets and sunup, rainbows, and certain plants and animals |
|
|
12. Simulating natural light and air |
- |
|
|
13. Naturalistic shapes and forms |
the shapes of plants on building facades and columns, animal facsimiles woven into fabrics and coverings. |
|
|
14. Evoking nature |
draw from design principles and characteristics of the natural world |
|
|
15. Information richness |
Rich sensory information |
|
|
16. Age, change, and the patina of time |
naturally aging materials, weathering, a sense of the passage of time |
|
|
17. Natural geometries |
hierarchically organized scales, sinuous rather than rigid artificial geometries, self-repeating but varying patterns |
|
|
18. Biomimicry |
forms and functions found in nature, especially among other species |
|
|
19. Prospect and refuge |
Prospect: long views of surrounding settings Refuge: sites of safety and security |
|
|
20. Organized complexity |
Complex spaces tend to be variable and diverse, while organized ones possess attributes of connection and coherence. |
|
|
21. Integration of parts to wholes |
sequential and successional linking of spaces, as well as by clear and discernible boundaries |
|
|
22. Transitional spaces |
hallways, thresholds, doorways, gateways, and areas that link the indoors and outdoors especially porches, patios, courtyards, colonnades |
|
|
23. Mobility and wayfinding |
Clearly understood pathways and points of entry and egress |
|
|
24. Cultural and ecological sites |
Local landscapes, indigenous flora and fauna, and characteristic meteorological conditions |
|
|
Column B: 14 Patterns of Biophilic Design (Browning et al., 2014) |
1. Visual Connection with nature |
A view to elements of nature, living systems and natural processes. |
|
2. Non-Visual Connection with nature |
Auditory, haptic, olfactory, or gustatory stimuli that engender a deliberate and positive reference to nature, living systems or natural processes. |
|
|
3. Non-Rhythmic Sensory Stimuli |
Stochastic and ephemeral connections with nature that may be analyzed statistically but may not be predicted precisely. |
|
|
4. Thermal & Air flow variability |
Subtle changes in air temperature, relative humidity, air ow across the skin, and surface temperatures that mimic natural environments. |
|
|
5. Presence of Water |
A condition that enhances the experience of a place through the seeing, hearing, or touching of water. |
|
|
6. Dynamic & Diffuse Light |
Leveraging varying intensities of light and shadow that change over time to create conditions that occur in nature. |
|
|
7. Connection with natural Systems |
Awareness of natural processes, especially seasonal and temporal changes characteristic of a healthy ecosystem. |
|
|
8. Biomorphic Forms & Patterns |
Symbolic references to contoured, patterned, textured or numerical arrangements that persist in nature. |
|
|
9. material Connection with nature |
Material and elements from nature that, through minimal processing, reflect the local ecology or geology to create a distinct sense of place. |
|
|
10. Complexity & Order |
adheres to a spatial hierarchy that similar to those encountered in nature |
|
|
11. Prospect |
An unimpeded view over a distance for surveillance and planning. |
|
|
12. Refuge |
A place for withdrawal, from environmental conditions or the main flow of activity, in which the individual is protected from behind and overhead. |
|
|
13. Mystery |
The promise of more information achieved through partially obscured views or other sensory devices that entice the individual to travel deeper into the environment. |
|
|
14. Risk/Peril |
An identifiable able threat coupled with a reliable safeguard. |
Second, we fill the correlated boxes with same solid colours to make them more recognizable:
Revision in Page 6 Line 207:
Point 11: Table 2: Please consider providing further explanations for this table. It was not clear what the authors wanted to show.
Response 11: Table 2 made a comparison between the selected nine biophilic design patterns for workplace in this study and the validated eight factors that affect workers’ satisfaction and productivity. There are overlaps between the nine biophilic design attributes and these eight influential factors for the workplace. These overlapped factors highlight the nine biophilic design attributes that are critical to the office design. Hence, the validation of the selection of the nine biophilic attributes are proofed by the previous literature. Sentences are added in the paragraph to clarify the demonstration.
Revision in Page 6 Line 215-220:
“Table 2 made a comparison between the selected nine biophilic design attributes for workplace in this study and the validated eight factors that affect workers’ satisfaction and productivity. There are overlaps between the nine biophilic design attributes and these eight influential factors for the workplace. These overlapped factors highlight the nine biophilic design at-tributes that are critical to the office design. The validation of the selection of the nine biophilic attributes are proofed by the previous literature [37].”
Point 12: P8, L245: Please specify why these two offices were of interest (e.g., were they comparable or had specify architectural features worthy of study). If possible, please provide more characteristics (e.g., size, floor area, furniture layout (e.g., open-plan or enclosed), etc.) for each office. Later (P10, L299), it says 201 questionnaires were collected, with 161 occupants taking part in the Singaporean office. This led me to believe that this office was much larger than the building studied in China. An image showing the indoor conditions and outdoor façade for each might be beneficial. Many of the explanations found in section 3.1 could be moved into this part of the manuscript, since they many describe and show the existing office conditions and to do necessarily form part of the main results.
Response 12: 1) The reasons why the two offices are selected for investigation: a) the urban contexts are similar: both the cities (Singapore and Shenzhen) are typical compact, high density Asian mega-cities; b) both the offices are open-plan offices.
Revision in Page 9 Line 277-280:
“The two offices have similar features: 1) the urban contexts are similar: both the cities (Singapore and Shenzhen) are typical compact, high density Asian mega-cities; b) both the offices are open-plan offices.”
2) The supplemental Information (i.e., temperature, number of employees) are added in the revised version.
Revision in Page 9 Table 5:
|
Dimension |
Office A |
Office B |
|
Location |
Singapore |
Shenzhen, South China |
|
Climate Zone |
Tropical Monsoon climate |
Sub-tropical climate |
|
Coordinate |
1°16′North, 103°5′East |
22°55′North, 114.1°East |
|
Floor |
8 |
10 |
|
Office Ventilation Type |
Central air conditioned |
Natural ventilation |
|
Temperature in the office |
25 to 26° C |
26 to 28°C |
|
No. of Employees |
approx. 300 |
approx. 150 |
Table 5. Basic information of the understudy offices.
- The structure is re-constructed in the revised version. The original Section 3.1. Observation Results-Biophilic Design Attributes in the Selected Offices is moved to be Section 2.3.2. It is because the 2.3.1 is the selection of office, and the Section 2.3.2 provides observation details of the selected offices. After re-construction, Section 3. Questionnaire Results focus on illustrating the questionnaire results.
Point 13: Section 3.3. Although I appreciated the thoroughness to which the descriptive statistical was explained, I wasn’t convinced the mean was the best indicator for the data, considering that evaluation scores were collected on a 5-point scale and not a continuous linear one. In-lieu of the mean, please consider using the median and inter-quartile range as the central tendency and dispersion indicators. Figures 1 and 4 can be removed, as the assumption of normality no longer applies (also on P15, L399-400), or replaced with boxplots.
Response 13: We agree with the comment. The means (SD) are only applied when the assumption of normality applies for the datasets. Hence, all the means (SD) are replaced as medians (IQR). And the corresponding figures (Figure 1 in Page 13 Line 359-361 and Figure 4 in Page 16-17 of the original manuscript) are removed in the revised version.
Revision in Page 13 Line 381-394:
The Cronbach’s α coefficient value of the main scale is 0.72, while those of the subscales GH, NR, and BDE in order are obtained as 0.68, 0.79, and 0.63, indicating that the questionnaire is reliable (i.e., an acceptable reliability: Cronbach's α>0.6) [57, 58] (Table 8).
According to the quantitative results presented in Table 8, the medians (Interquartile Range, IQR) of the assessment show moderately high opinions toward the health and wellbeing of biophilic offices (HWBO), at 71.00 (8.00) (the score range from min. 20 to max. 100). Concerning the self-related evaluation scales GH and NR, the score range of GH and HR are minimum 4 to maximum 20, and the obtained results illustrate moderately high opinions, the values of 15.00 (2.00) and 14.00 (3.00) for GH and NR. The median (IQR) value of the POE scale BDE is evaluated as 42.00 (5.00) (range of total value: min. 12 to max. 60).
Table 8. Medians, Interquartile Range (IQR), Mean, standard deviation, and α coefficient values of workers’ evaluation based on HWBO, GH, NR, and BDE.
|
Structure (item) |
Scale |
Median (IQR) |
Cronbach’s α |
|
Main scale (1) |
Health and Wellbeing of Biophilic Offices (HWBO) |
71.00 (8.00) |
.72 |
|
Subscales of main scale (3) |
General Health (GH) |
15.00 (2.00) |
.68 |
|
Nature Relatedness (NR) |
14.00 (3.00) |
.79 |
|
|
Biophilic Design Evaluation (BDE) |
42.00 (5.00) |
.63 |
Revision in Page 18-20 Table 9:
Table 9. Comparison of independent variables (Gender, Age, Educational Levels, Weekly Work Hours, Daily Sedentary Time, Work Desk Locations, Working Years, Office Locations) on self-reported GH, NR, and BDE.
|
Survey measures |
Category |
GH |
NR |
BDE |
|||
|
Median (IQR) |
Sig. |
Median (IQR) |
Sig. |
Median (IQR) |
Sig. |
||
|
Gender |
Male |
15.00 (2.00) |
.024* |
15.00 (2.00) |
.000* |
42.00 (5.00) |
.950 |
|
female |
15.00 (3.00) |
14.00 (3.00) |
42.00 (6.00) |
||||
|
Age |
21-25 |
16.00 (3.00) |
.009* |
15.00 (3.00) |
.004* |
43.00 (4.00) |
.115 |
|
26-35 |
15.00 (2.00) |
15.00 (3.00) |
42.00 (5.00) |
||||
|
36-45 |
15.00 (4.00) |
13.50 (3.00) |
41.00 (6.00) |
||||
|
46-60 |
15.50 (2.00) |
15.00 (2.00) |
40.50 (6.00) |
||||
|
Educational level |
Secondary school or equivalent |
14.00 (4.00) |
.384 |
14.00 (2.00) |
.656 |
39.00 (6.00) |
.391 |
|
Diploma or college certificate |
15.00 (2.00) |
15.00 (3.00) |
42.00 (5.00) |
||||
|
Master's degree or above |
15.00 (2.00) |
14.00 (3.00) |
42.00 (5.00) |
||||
|
Weekly work hours |
30 to 40 hours |
14.00 (2.00) |
.742 |
14.00 (3.00) |
.445 |
41.00 (6.00) |
.919 |
|
41 to 50 hours |
15.00 (2.00) |
14.00 (3.00) |
42.00 (5.00) |
||||
|
Above 50 hours |
15.00 (3.00) |
15.00 (3.00) |
42.00 (5.00) |
||||
|
Daily sedentary time |
Within 30 min |
15.00 (2.00) |
.549 |
15.00 (2.00) |
.216 |
42.00 (5.00) |
.652 |
|
30 min to 2 hours |
14.00 (3.00) |
13.00 (4.00) |
43.00 (5.00) |
||||
|
2 to 5 hours |
15.00 (2.00) |
14.00 (4.00) |
42.00 (5.00) |
||||
|
5 to 8 hours |
15.00 (2.00) |
15.00 (3.00) |
41.00 (5.00) |
||||
|
Above 8 hours |
15.00 (2.00) |
14.50 (4.00) |
41.00 (7.00) |
||||
|
Work desk location |
Window seats with natural views |
15.00 (3.00) |
.751 |
14.00 (3.00) |
.228 |
42.00 (5.00) |
.703 |
|
Window seats with urban views |
16.00 (2.00) |
14.00 (1.00) |
42.00 (5.00) |
||||
|
Aisle seats without window view |
15.00 (2.00) |
14.00 (3.00) |
42.00 (5.00) |
||||
|
Working years (in this company) |
1 year or less |
15.00 (2.00) |
.218 |
15.00 (3.00) |
.531 |
42.00 (5.00) |
.458 |
|
1-3 years |
15.00 (3.00) |
14.00 (4.00) |
42.00 (7.00) |
||||
|
3-5 years |
15.00 (3.00) |
14.50 (3.00) |
42.00 (6.00) |
||||
|
Above 5 years |
15.00 (2.00) |
14.00 (3.00) |
41.00 (6.00) |
||||
|
Office location |
Singapore |
15.00 (2.00) |
.995 |
14.00 (3.00) |
.701 |
41.00 (5.00) |
.244 |
|
Shenzhen, China |
15.00 (3.00) |
15.00 (3.00) |
43.00 (5.00) |
||||
Point 14: Table 8: Please consider applying benchmarks for what constitute reasonable levels for internal consistency, when using the Cronbach’s Alpha (e.g., α>0.7): Please see, for example: Taber, 2018. The use of Cronbach’s Alpha when developing and reporting research instruments in science education. Research in Science Education.
Tavakol et al. Making sense of Cronbach’s Alpha. International Journal of Medical Education.
Response 14: Thank you for your suggestion. The Cronbach’s Alpha with an Alpha>0.6 considered acceptable internal consistency in this study. The statement and the references are added in the revised version.
Revision in Page 13 Line 381-383:
“The Cronbach’s α coefficient value of the main scale is 0.72, while those of the sub-scales GH, NR, and BDE in order are obtained as 0.68, 0.79, and 0.63, indicating that the questionnaire is reliable (i.e., an acceptable reliability: Cronbach's α>0.6) [57, 58] (Table 8).”
Citations are added in the Reference list:
- Morgan, P. J., Cleave‐Hogg, D., DeSousa, S., Tarshis, J., 2004. High‐fidelity patient simulation: validation of performance checklists. British Journal of Anesthesia, Volume 92, (3) 388–392.
- Cronbach, L.J., 1951. Coefficient Alpha and the Internal Structure of tests, Psychometrika. Vol. 6. 3. 297-334.
Point 15: Figure 3. The plot is well presented. A few minor notes for improvement: 1) Please consider adding short or abbreviated labels referring to the actual question, instead of codes (e.g., GH3-Q10). This would make it easier for the reader; 2) Round the percentages to the nearest whole number.
Response 15: 1) Abbreviated labels referring to the actual questions are added in the Figure 3 (Figure 2 in the revision). But 2) we keep the original percentages (round to one decimal place). We think it is not hard for the readers to understand, and the original percentages (round to one decimal place) is more accurate.
Revision in Page 17 Line 431:
Point 16: P17, L434: Please correct the unfortunate citation error on this line.
Response 16: Thanks for the reminder, the error has been corrected in the revised version.
Revision in Page 21 Line 473:
“According to Table 10, Pearson correlations indicate that…”
Point 17: P18, section 4.2: Similar to comment #13, the data may be more suited to a Spearman’s correlation test, instead of the Pearson’s correlation coefficient. Due to the reasonable size of the dataset collected, it may not change the interpretation, but would help improve the analytical rigor.
Response 17: Spearman’s correlation test was conducted, and the revised contents are shown in the updated version.
Revision in Page 17 Line 440:
“Second, Spearman's correlation analysis is utilized to examine the correlation between three subscales.”
Revision in Page 21 Line 472-483:
4.2. Intercorrelation between the three subscales (GH, NR, BDE)
“According to Table 10, Spearman's correlations indicate that works' nature relatedness (NR) was positively correlated with self-evaluated GH (r = .264**, p < .01). This result also confirms the previously obtained results that people who had a higher evaluation in nature relatedness are also had a higher evaluation on their health. When the occupants feel that they have a strong sense of relationship with nature, it is observed that the biophilic environment would have positive impacts on their health. More importantly, significant correlation is also found in between biophilic design evaluation and self-reported health (GH), r=.270**, p < .01, indicating that office biophilic design has positive values on workers’ psychological health.”
Table 10. Intercorrelations between responses of three subscales.
|
|
GH |
NR |
BDE |
|
General Health |
- |
.264** |
.270** |
|
Nature Relatedness |
.264** |
- |
.135 |
|
Biophilic Design Evaluation |
.270** |
.135 |
- |
**. Correlation is significant at the 0.01 level (2-tailed)
Point 18: P18, L448-450: Please check whether the sentence is accurate and correct the table caption numbers; I believe these should be Tables 10 and 11 and 12, not 1, 2 and 3. The sentence reads: Homogenous subsets with significant discrepancies (differences?) across subsets, leading to no significant differences across subsets. The above is not easy to grasp. If the information is accurate, please consider amending this to make this clearer.
Response 18: Thank you for your comment. The above contents have been corrected in the revised version.
Point 19: While the conclusions were well structured, I felt the authors could have highlighted more the main takeaway messages from their endeavors, in particularly the relationship between biophilic design and occupant health and wellbeing. This seems to be a core aspect of their work but did really emerge from the final section of their work in the same way it was emphasized in the abstract.
Response 19: The Conclusion is rewritten to highlight the relationship between biophilic design and occupant health and wellbeing:
Revision in Page 23 Line 514-540:
“The significant research outputs from the present scrutiny are shown as following:
- a) The authors develop a POE questionnaire for evaluating the biophilic design for workplace health and wellbeing. The investigation explains that combined literature review and POE results are one of the practical methodologies to establish biophilic design frameworks for specific workplace typology. And the questionnaire can be applied in future biophilic design research for investigation.
- b) Additionally, the study provides novel design guidelines for designers with emphasizing on weight for workplace design practices. The weighting results of this study would be especially applicable to the workplace typology. The 14 Patterns of Biophilic Design has a broader range of usage for all building typologies and is more suitable for general design applications. Therefore, the weighting results of this experiment are not employed to deny the ranking in the 14 Patterns of Biophilic Design. These are exploited to show a new biophilic design framework for the workplace according to the users’ points of view (based on the POE results).
- c) Furthermore, the questionnaire results enhance our knowledge on the practical ap-plication of biophilic design frameworks for the workplace and contributed to more framework design consideration.
- d) The correlation results support the importance of biophilic design from the user perspectives. There is a significant correlation between office biophilic design and self-reported health of employees (r=.270**, p < .01).
- e) The study results contribute to provide designers with evidence-based design at-tributes for workplace design (i.e., the nine selected workplace biophilic design attributes).”

Reviewer 2 Report
The study about the evaluation of biophilic attributes in the workplace for improving health and wellbeing is actual and interesting and therefore the paper is fulfilling the scientific criteria in order to be published. Practical methodologies to establish biophilic design frameworks for specific workplace typology
Author Response
Point 1: The study about the evaluation of biophilic attributes in the workplace for improving health and wellbeing is actual and interesting and therefore the paper is fulfilling the scientific criteria to be published. Practical methodologies to establish biophilic design frameworks for specific workplace typology.
Response 1: Thank you for your comments and suggestions.

Reviewer 3 Report
Comments for buildings-1613867.
The paper discusses the impacts of biophilic design attributes in offices on workers’ health and wellbeing. The paper is clear and informative. In this paper, through two practical study some results of persuasive and constructive significance. However, some parts of the article which can be optimized.
1.The introduction can be optimized appropriately.The introduction of the term "Biophilia" can be more detailed and easy to understand .By contrast, the introduction of “Biophilic Design”is very substantial.It may be better to find some literature to combine workplace design and “Biophilic Design”,and then to explore the relationship between them.
2.The illustrations in the article are small and a bit vague,some pictures can shrink a little, not to the top to the border,these can be optimized.
3.Overall work of the article is sufficient, the results is has the certain significance, but its limitation is obvious to all. Some of the results in the illustrations are obvious and can be reduced to less elaboration.
4.The result of biophilic design frameworks for specific workplace typology is of certain value. The location of the workplace selected in this article is limited, the related biophilic factors are scientifically screened, and the researchers have made a detailed classification study. However, the huge research scope has certain obstacles to the relevant results, and the general and targeted conclusions will be worse.
- It is best to supplement and describe the necessity of research. At the same time, it will be more complete if the results are reflected in the summary.
- The object of the questionnaire should be composed of people with different ages and genders, which is best explained in the article.
- Adding some explanations, examples and data comparison to the conclusion will be more intuitive and convincing, just as discussed earlier in the article. Perhaps this makes this article more complete and credible. At the same time, the conclusion only summarizes the article, and lasks discussions and explanations for the future research direction.
- The theoretical part of the presentation is extensive, but the application part is relatively poorly described
- It is better to supplement the temperature, humidity and other parameters of the selected office in the part of the experiment, so as to facilitate readers' reference rather than just giving the location.
- The author can think about the impact of such a pro biological design model on the psychology of different experimental personnel. I think psychological factors will also affect human physiological comfort.
Author Response
Point 1:
The introduction can be optimized appropriately. The introduction of the term "Biophilia" can be more detailed and easier to understand. By contrast, the introduction of “Biophilic Design” is very substantial. It may be better to find some literature to combine workplace design and “Biophilic Design”, and then to explore the relationship between them.
Response 1:
1) Further explanations of “Biophilia” are added in the Introduction.
Revision in Page 1 Line 27-29:
“The term "Biophilia" is evolved from human evolution research and is coined to describe humans' inherent love affinity for the living things in the natural world [1,2]. It explained why we prefer nature because it is an instinct deeply rooted in the human brain.”
2) Literatures are added in the revised version to combine “workplace design” and “Biophilic Design”.
Revision in Page 2 Line 48-53:
“Workplace is one of the typologies that attracts the attentions of researchers. Scholars who research the relationship between the built environment and health found that the environment not merely directly or indirectly affects human health but also affects their work and study performance [59]. Studies proofed that biophilic design benefits workers’ health and productivity in an office environment [60, 61, 62, 63].”
Citations are added in the Reference list:
- Derek, C. C., 2003. Environmental Quality and the Productive Workplace. In C. J. Anumba (Ed.), Innovative Developments in Architecture, Engineering and Construction. Rotterdam: Millpress Science Publishers.
- Lei, Q.H., Yuan, C., Lau, S.S.Y., 2021. A quantitative study for indoor workplace biophilic design to improve health and productivity performance. Journal of Cleaner Production. 324, 129168.
- Yin, J., Zhua, S., MacNaughton, P., Joseph, G., Allen, J.G., Spengler, J.D., 2018. Physiological and cognitive performance of exposure to biophilic indoor environment. Build. Environ. 132, 255–262.
- Yin, J., Arfaei, N., MacNaughton, P., Catalano, P.J., Allen, J.G., Spengler, J.D., 2019. Effects of biophilic interventions in office on stress reaction and cognitive function: a randomized crossover study in virtual reality. Indoor Air 29, 1028–1039. https:// doi.org/10.1111/ina.12593.
- Yin, J., Yuan, J., Arfaei, N., Catalano, P.J., Allen, G.J., Spengler, J.D., 2020. Effects of biophilic indoor environment on stress and anxiety recovery: a between-subjects experiment in virtual reality. Environ. Int. 136, 105427. https://doi.org/10.1016/j. envint.2019.105427.
Point 2:
The illustrations in the article are small and a bit vague, some pictures can shrink a little, not to the top to the border, these can be optimized.
Response 2:
Thanks for your suggestion. The figures and pictures have been adjusted in the revised manuscript.
Point 3:
Overall work of the article is sufficient, the results have the certain significance, but its limitation is obvious to all. Some of the results in the illustrations are obvious and can be reduced to less elaboration.
Response 3:
Thanks for your suggestion. The figures and pictures have been adjusted in the revised manuscript.
The Section 3 Questionnaire Results and Section 4 are rewritten in the revised version: a) the statistical analysis is modified; b) to reduce elaboration in the text. Please kindly refer the following revisions:
1) The original Section 3.1. Observation Results-Biophilic Design Attributes in the Selected Offices is moved to be Section 2.3.2. It is because the 2.3.1 is the selection of office, and the Section 2.3.2 provides observation details of the selected offices. After re-construction, Section 3. Questionnaire Results focus on illustrating the questionnaire results.
2) In the revised version, the formulars in Section 3.3 2. Quantitative Results of Impacts of Biophilic Design for Workplace are removed due to the statistical analysis is modified:
The means (SD) are only applied when the assumption of normality applies for the datasets. Hence, all the means (SD) are replaced as medians (IQR). And the corresponding figures (Figure 1 in Page 13 Line 359-361 and Figure 4 in Page 16-17 of the original manuscript) are removed in the revised version.
Revision in Page 13 Line 381-394:
The Cronbach’s α coefficient value of the main scale is 0.72, while those of the subscales GH, NR, and BDE in order are obtained as 0.68, 0.79, and 0.63, indicating that the questionnaire is reliable (i.e., an acceptable reliability: Cronbach's α>0.6) [57, 58] (Table 8).
According to the quantitative results presented in Table 8, the medians (Interquartile Range, IQR) of the assessment show moderately high opinions toward the health and wellbeing of biophilic offices (HWBO), at 71.00 (8.00) (the score range from min. 20 to max. 100). Concerning the self-related evaluation scales GH and NR, the score range of GH and HR are minimum 4 to maximum 20, and the obtained results illustrate moderately high opinions, the values of 15.00 (2.00) and 14.00 (3.00) for GH and NR. The median (IQR) value of the POE scale BDE is evaluated as 42.00 (5.00) (range of total value: min. 12 to max. 60).
Table 8. Medians, Interquartile Range (IQR), Mean, standard deviation, and α coefficient values of workers’ evaluation based on HWBO, GH, NR, and BDE.
|
Structure (item) |
Scale |
Median (IQR) |
Cronbach’s α |
|
Main scale (1) |
Health and Wellbeing of Biophilic Offices (HWBO) |
71.00 (8.00) |
.72 |
|
Subscales of main scale (3) |
General Health (GH) |
15.00 (2.00) |
.68 |
|
Nature Relatedness (NR) |
14.00 (3.00) |
.79 |
|
|
Biophilic Design Evaluation (BDE) |
42.00 (5.00) |
.63 |
Revision in Page 18 Table 9:
Table 9. Comparison of independent variables (Gender, Age, Educational Levels, Weekly Work Hours, Daily Sedentary Time, Work Desk Locations, Working Years, Office Locations) on self-reported GH, NR, and BDE.
|
Survey measures |
Category |
GH |
NR |
BDE |
|||
|
Median (IQR) |
Sig. |
Median (IQR) |
Sig. |
Median (IQR) |
Sig. |
||
|
Gender |
Male |
15.00 (2.00) |
.024* |
15.00 (2.00) |
.000* |
42.00 (5.00) |
.950 |
|
female |
15.00 (3.00) |
14.00 (3.00) |
42.00 (6.00) |
||||
|
Age |
21-25 |
16.00 (3.00) |
.009* |
15.00 (3.00) |
.004* |
43.00 (4.00) |
.115 |
|
26-35 |
15.00 (2.00) |
15.00 (3.00) |
42.00 (5.00) |
||||
|
36-45 |
15.00 (4.00) |
13.50 (3.00) |
41.00 (6.00) |
||||
|
46-60 |
15.50 (2.00) |
15.00 (2.00) |
40.50 (6.00) |
||||
|
Educational level |
Secondary school or equivalent |
14.00 (4.00) |
.384 |
14.00 (2.00) |
.656 |
39.00 (6.00) |
.391 |
|
Diploma or college certificate |
15.00 (2.00) |
15.00 (3.00) |
42.00 (5.00) |
||||
|
Master's degree or above |
15.00 (2.00) |
14.00 (3.00) |
42.00 (5.00) |
||||
|
Weekly work hours |
30 to 40 hours |
14.00 (2.00) |
.742 |
14.00 (3.00) |
.445 |
41.00 (6.00) |
.919 |
|
41 to 50 hours |
15.00 (2.00) |
14.00 (3.00) |
42.00 (5.00) |
||||
|
Above 50 hours |
15.00 (3.00) |
15.00 (3.00) |
42.00 (5.00) |
||||
|
Daily sedentary time |
Within 30 min |
15.00 (2.00) |
.549 |
15.00 (2.00) |
.216 |
42.00 (5.00) |
.652 |
|
30 min to 2 hours |
14.00 (3.00) |
13.00 (4.00) |
43.00 (5.00) |
||||
|
2 to 5 hours |
15.00 (2.00) |
14.00 (4.00) |
42.00 (5.00) |
||||
|
5 to 8 hours |
15.00 (2.00) |
15.00 (3.00) |
41.00 (5.00) |
||||
|
Above 8 hours |
15.00 (2.00) |
14.50 (4.00) |
41.00 (7.00) |
||||
|
Work desk location |
Window seats with natural views |
15.00 (3.00) |
.751 |
14.00 (3.00) |
.228 |
42.00 (5.00) |
.703 |
|
Window seats with urban views |
16.00 (2.00) |
14.00 (1.00) |
42.00 (5.00) |
||||
|
Aisle seats without window view |
15.00 (2.00) |
14.00 (3.00) |
42.00 (5.00) |
||||
|
Working years (in this company) |
1 year or less |
15.00 (2.00) |
.218 |
15.00 (3.00) |
.531 |
42.00 (5.00) |
.458 |
|
1-3 years |
15.00 (3.00) |
14.00 (4.00) |
42.00 (7.00) |
||||
|
3-5 years |
15.00 (3.00) |
14.50 (3.00) |
42.00 (6.00) |
||||
|
Above 5 years |
15.00 (2.00) |
14.00 (3.00) |
41.00 (6.00) |
||||
|
Office location |
Singapore |
15.00 (2.00) |
.995 |
14.00 (3.00) |
.701 |
41.00 (5.00) |
.244 |
|
Shenzhen, China |
15.00 (3.00) |
15.00 (3.00) |
43.00 (5.00) |
||||
*p<.05, The significance level is .05.
3) Elaboration for the Figure 2. Stacked graph with percentage responses for individually arranged items (from the bottom with a high percentage of disagreement to the top with high percentage of agreement) is reduced, because the illustrations are obvious in the figure.
Revision in Page 15-17 Line 399-433:
The analysis of individual items provides more details into the works' responses. As can be seen in percentage responses for individual items (Figure 2 in revision), most of the responses are distributed in the items "Neutral" and "Agree". The questionnaire results reveal that the employees from the understudy companies hold a relatively positive opinion on wellbeing, nature-relatedness, indoor environmental quality, and biophilic design for their health promotion. According to the arrangement, at the top of this, stacked graph are the evaluation of satisfaction of the work capacities and relationships in the workplace. About 73.2% of respondents agree that the workers of the companies under investigation are satisfied with their work capacity (GH3-Q10) and relationships (GH4-Q11). In the subscale nature relatedness, 62.2% and 61.7% of workers responded (agree/strongly agree) that their actions affect the environment (NR2-Q13), and they take notice of the wildlife in their daily lives (NR3-Q14). Nevertheless, only 47.8% of them selected agreed/strongly agree regarding the statements that their ideal spot for vacation would be a wilderness area (NR1-Q12). Regarding the POE results in the subscale BDE, 63.2% of workers agree/strongly agree that the natural light is an essential biophilic attribute and their offices are bright (BDE2-Q17). Furthermore, 60.7 % of the workers agreed that introducing natural colors in the office benefits workplace health and wellbeing (BDE11-Q26). More than 60 percent (approximately 60.7%) of respondents believe that greenery is a biophilic design that benefits office wellbeing (BDE6-Q21). Their feedback would be valuable for designers to note that application of the biophilic design attributes in the office design can enhance the experiences and evaluations of workers.
The quantitative results of the questionnaire survey demonstrate that the workers agree that the biophilic design attributes in the office have positive effects on their subjective wellbeing.
Figure 2. Stacked graph with percentage responses for individually arranged items (from the bottom with a high percentage of disagreement to the top with high percentage of agreement).
Point 4:
The result of biophilic design frameworks for specific workplace typology is of certain value. The location of the workplace selected in this article is limited, the related biophilic factors are scientifically screened, and the researchers have made a detailed classification study. However, the huge research scope has certain obstacles to the relevant results, and the general and targeted conclusions will be worse.
Response 4:
Thank you for your comments. These considerations are the limitations of this study. 1) these two cases are limited in representing all the workplace biophilic designs. As mentioned in the Conclusion, in the future study, we will include more offices and locations as experiment samples. 2) The investigative POE studies evaluated the self-reported health (GH), nature-relatedness (NR), and biophilic design in the workplace (BDE). The objective of the study is to evaluate the typical biophilic design attributes in office environment and the correlation between biophilic design and office health. Hence, the research scope is relatively extensive. In the future study, the research scope should be narrow down for intensive investigation.
Point 5:
It is best to supplement and describe the necessity of research. At the same time, it will be more complete if the results are reflected in the summary.
Response 5:
Thank you for your suggestion. The description of the necessity of research is highlight in the revised version. Please kindly refer the following revisions:
Revision in Page 2 Line 56-66:
“Although the importance of biophilic design seems to be well-acknowledged, and some international or regional green building and healthy building standards incorporate biophilic design elements into the rating system, such as WELL building standard version 2 and Singapore Green Mark [53]. However, further research on developing building typology-based biophilic design guide-lines and assessment methods are necessary. Additionally, the effectiveness of such design in practical design projects for user wellbeing still requires confirmation. More importantly, building typology-based biophilic design guidelines should be appropriately developed because it would affect the designer's prioritization of design attributes selection in design practice.”
Point 6:
The object of the questionnaire should be composed of people with different ages and genders, which is best explained in the article.
Response 6:
The different genders and different ages are included in the study. The age ranges included 21-25, 26-35, 36-45, 46-60. The detailed description of the demographic information is in Section 3.1. Demographic Information.
Point 7:
Adding some explanations, examples and data comparison to the conclusion will be more intuitive and convincing, just as discussed earlier in the article. Perhaps this makes this article completer and more credible. At the same time, the conclusion only summarizes the article, and lacks discussions and explanations for the future research direction.
Response 7:
1) The Conclusion is rewritten to highlight the relationship between biophilic design and occupant health and wellbeing:
Revision in Page 23 Line 516-540:
“The significant research outputs from the present scrutiny are shown as following:
- a) The authors develop a POE questionnaire for evaluating the biophilic design for workplace health and wellbeing. The investigation explains that combined literature review and POE results are one of the practical methodologies to establish biophilic design frameworks for specific workplace typology. And the questionnaire can be applied in future biophilic design research for investigation.
- b) Additionally, the study provides novel design guidelines for designers with emphasizing on weight for workplace design practices. The weighting results of this study would be especially applicable to the workplace typology. The 14 Patterns of Biophilic Design has a broader range of usage for all building typologies and is more suitable for general design applications. Therefore, the weighting results of this experiment are not employed to deny the ranking in the 14 Patterns of Biophilic Design. These are exploited to show a new biophilic design framework for the workplace according to the users’ points of view (based on the POE results).
- c) Furthermore, the questionnaire results enhance our knowledge on the practical ap-plication of biophilic design frameworks for the workplace and contributed to more framework design consideration.
- d) The correlation results support the importance of biophilic design from the user perspectives. There is a significant correlation between office biophilic design and self-reported health of employees (r=.270**, p < .01).
- e) The study results contribute to provide designers with evidence-based design at-tributes for workplace design (i.e., the nine selected workplace biophilic design attributes).”
2) The Conclusion is reconstructed and a separate section-- Section 6. Limitation and Future Studies is added in the revised version:
Revision in Page 23 Line 542-551:
“6. Limitation and Future Studies
There are limitations in this study, first, these two cases are limited in representing all the workplace biophilic designs due to the sample size limitations. Further studies could include more offices and locations as experiment samples. 2) The investigative POE studies evaluated the self-reported health (GH), nature-relatedness (NR), and biophilic design in the workplace (BDE). The objective of the study is to evaluate the typical biophilic design attributes in office environment and the correlation between biophilic design and office health. Hence, the research scope is relatively extensive. In the future study, the research scope should be narrow down for intensive investigation.”
Point 8:
The theoretical part of the presentation is extensive, but the application part is relatively poorly described.
Response 8:
The applications of the results are highlighted in Section 1.4. Objectives and in Section 5 Conclusion:
In Section 1.4. Objectives, the three practical applications of the results are mentioned: first, a POE questionnaire for assessing the biophilic design for workplace accounting for health and wellbeing. Second, the study will provide a new biophilic design guidelines for workplaces, which can effectively assist researchers and designers to improve office biophilic design practices and decision-making on design attributes selection.
The Conclusion is rewritten to highlight the contribution and implementation of the results:
Revision in Page 23 Line 516-540:
“The significant research outputs from the present scrutiny are shown as following:
- a) The authors develop a POE questionnaire for evaluating the biophilic design for workplace health and wellbeing. The investigation explains that combined literature review and POE results are one of the practical methodologies to establish biophilic design frameworks for specific workplace typology. And the questionnaire can be applied in future biophilic design research for investigation.
…
“c) Furthermore, the questionnaire results enhance our knowledge on the practical ap-plication of biophilic design frameworks for the workplace and contributed to more framework design consideration.”
…
“e) The study results contribute to provide designers with evidence-based design at-tributes for workplace design (i.e., the nine selected workplace biophilic design attributes).”
Point 9:
It is better to supplement the temperature, humidity, and other parameters of the selected office in the part of the experiment, so as to facilitate readers' reference rather than just giving the location.
Response 9:
The supplemental Information (i.e., office temperature, number of employees) are added in the revised version.
Revision in Table 5:
Table 5. Basic information of the understudy offices.
|
Dimension |
Office A |
Office B |
|
Location |
Singapore |
Shenzhen, South China |
|
Climate Zone |
Tropical Monsoon climate |
Sub-tropical climate |
|
Coordinate |
1°16′North, 103°5′East |
22°55′North, 114.1°East |
|
Floor |
8 |
10 |
|
Office Ventilation Type |
Central air conditioned |
Natural ventilation |
|
Temperature in the office |
25 to 26° C |
26 to 28°C |
|
No. of Employees |
approx. 300 |
approx. 150 |
Point 10:
The author can think about the impact of such a pro biological design model on the psychology of different experimental personnel. I think psychological factors will also affect human physiological comfort.
Response 10:
We agree with the comment. Previous studies proofed that biophilic design in workplaces benefits both psychological and physiological health [60, 61, 62, 63]. This study is a questionnaire survey that focus on investigating the subjective evaluation on workplace biophilic design and of workers. Physiological measurements will be included in the future study to investigate the impacts of the psychological factors on physiological comfort.
References:
- Lei, Q.H., Yuan, C., Lau, S.S.Y., 2021. A quantitative study for indoor workplace biophilic design to improve health and productivity performance. Journal of Cleaner Production. 324, 129168.
- Yin, J., Zhua, S., MacNaughton, P., Joseph, G., Allen, J.G., Spengler, J.D., 2018. Physiological and cognitive performance of exposure to biophilic indoor environment. Build. Environ. 132, 255–262.
- Yin, J., Arfaei, N., MacNaughton, P., Catalano, P.J., Allen, J.G., Spengler, J.D., 2019. Effects of biophilic interventions in office on stress reaction and cognitive function: a randomized crossover study in virtual reality. Indoor Air 29, 1028–1039. https:// doi.org/10.1111/ina.12593.
- Yin, J., Yuan, J., Arfaei, N., Catalano, P.J., Allen, G.J., Spengler, J.D., 2020. Effects of biophilic indoor environment on stress and anxiety recovery: a between-subjects experiment in virtual reality. Environ. Int. 136, 105427. https://doi.org/10.1016/j. envint.2019.105427.

Round 2
Reviewer 1 Report
I would like to thank the authors for taking all my comments into full consideration. After reviewing their revised manuscript, I felt that their responses were on-point and addressed my earlier thoughts. I also appreciated the level of detail they provided for responses to #10 and #12. Figure 1, in particular, is now much easier to read and extract relevant information from in the graphical plot. In general, I felt that there was a significant improvement to their work.
There were two very minor points that I came across, which they may want to reconsider before proceeding with their article. These are not significant remarks, but nonetheless, might provide further improvements and clarity:
#1: For their response to comment #6, while I generally agreed with the authors, I felt that it didn’t quite address what I tried to originally convey. POE scales do not likely target biophilic elements in architectural designs directly, since a direct question (e.g. how satisfied are you with the biophilic features) may not accurately depict every beneficial nuance they offer (e.g. psychological recovery). Instead studies may elect to use, for example, other relatable scales, which may not have originally been designed for POE surveys (e.g. Mayer’s connectedness to nature scale) to help measure them. My generally feeling is that scales, from other domains (i.e. outside of POE studies) have been adopted for this reason, and a short sentence explaining the rationale supporting the lack of scales for biophilic design could be provided to briefly mentioned this.
#2: In consideration to their response #8, I understood and agreed with why half the seven of the 14 papers were discarded, but it did not provide much insight into why the Patterns of Biophilic Design were still used. A more useful example, which may have elucidated this more, was whether are there too few frameworks available, and/or none others could have been used. If this was the case, perhaps this could be succinctly mentioned.
Author Response
Response to Reviewer 1 Comments
I would like to thank the authors for taking all my comments into full consideration. After reviewing their revised manuscript, I felt that their responses were on-point and addressed my earlier thoughts. I also appreciated the level of detail they provided for responses to #10 and #12. Figure 1, in particular, is now much easier to read and extract relevant information from in the graphical plot. In general, I felt that there was a significant improvement to their work. There were two very minor points that I came across, which they may want to reconsider before proceeding with their article. These are not significant remarks, but nonetheless, might provide further improvements and clarity:
Point 1: For their response to comment #6, while I generally agreed with the authors, I felt that it didn’t quite address what I tried to originally convey. POE scales do not likely target biophilic elements in architectural designs directly, since a direct question (e.g., how satisfied are you with the biophilic features) may not accurately depict every beneficial nuance they offer (e.g., psychological recovery). Instead, studies may elect to use, for example, other relatable scales, which may not have originally been designed for POE surveys (e.g., Mayer’s connectedness to nature scale) to help measure them. My generally feeling is that scales, from other domains (i.e., outside of POE studies) have been adopted for this reason, and a short sentence explaining the rationale supporting the lack of scales for biophilic design could be provided to briefly mentioned this.
Response 1: Thank you for your comment. In this revised version, we further explain the lack of scales for biophilic design evaluation. Please find the revised sentences as below:
Revision in Page 3 Line 122-128:
“Actually, these are standard research methods of investigative POE; nevertheless, there are no existing POE scales that focus on biophilic design in the workplace. According to the research objectives of this study (i.e., evaluate the subjective health impacts of biophilic design in workplace), we need to refer the well-developed scales from other disciplinary (e.g., Environmental Psychology). And finally, a scale that integrated health evaluation and building environment evaluation (i.e., POE) is developed for investigation.”
Point 2: In consideration to their response #8, I understood and agreed with why half the seven of the 14 papers were discarded, but it did not provide much insight into why the Patterns of Biophilic
Design was still used. A more useful example, which may have elucidated this more, was whether: are there too few frameworks available, and/or no others could have been used. If this was the case, perhaps this could be succinctly mentioned.
Response 2: Thank you for your suggestions. In the revised version, sentences are added to further explain that there are too few framework available, and these two frameworks are the mainstream biophilic design frameworks that are widely applied in the practical biophilic design projects.
Revision in Page 3 Line 122-128:
“The 24 Biophilic Design Attributes and the 14 Patterns of Biophilic Design are the two mainstream biophilic design which are widely applied in the practical biophilic design projects. Hence, these two frameworks are the most suitable to be selected as the basis of this experiment.”
Reviewer 3 Report
1, Figure 1 should be improved regarding the arrow.
2, Photo in table 6 should check the copyright of these photo.
3, Please bring strong relevance to the scope of journal "Buildings" by investigating most recent literature.
Author Response
Response to Reviewer 3 Comments
Point 1: Figure 1 should be improved regarding the arrow.
Response 1: Thank you for your comment. Figure 1 is updated in this version.
Revision in Page 6 Line 216:
Point 2: Photo in table 6 should check the copyright of these photo.
Response 2: All the photos in Table 6 are taken by the designers. And we added an annotation under Table 6.
Revision in Page 11 Line 339:
“(Photos: by the authors)”
Point 3: Please bring strong relevance to the scope of journal "Buildings" by investigating most recent literature.
Response 3: Thank you for your suggestions. The references are added in the manuscript.
In-text citations in Page 1 Line 27-28:
“The term "Biophilia" is evolved from human evolution research and is coined to de-scribe humans' inherent love affinity for the living things in the natural world [1,2,64].”
In-text citations in Page 1 Line 40:
“…the modern built environment' as 'Biophilic Design' [24,25,65].”
In-text citations in Page 2 Line 69-70:
“…occupant satisfaction, health, and wellbeing after occupancy of buildings [27, 28, 67].”
Citations are added in the Reference list:
- Parsaee, M., Demers, M. H. C., Potvin, A., Hébert, M., Lalonde, J.F., Window View Access in Architecture: Spatial Visualization and Probability Evaluations Based on Human Vision Fields and Biophilia. 2021. Buildings, 11(12), 627. https://doi.org/10.3390/buildings11120627
- Mollazadeh, M., Zhu, YM., Application of Virtual Environments for Biophilic Design: A Critical Review. 2021. Buildings, 11(4), 148; https://doi.org/10.3390/buildings11040148
- Gillis, K., Gatersleben, B., 2015. A Review of Psychological Literature on the Health and Wellbeing Benefits of Biophilic Design. Buildings. 5(3), 948-963. https://doi.org/10.3390/buildings5030948
- Candido, C., Chakraborty, P., Tjondronegoro, D., 2019. The Rise of Office Design in High-Performance, Open-Plan Environments. Buildings. 9(4), 100. https://doi.org/10.3390/buildings9040100
